# Understanding Deep Contrastive Learning via Coordinate-wise Optimization

**Yuandong Tian**
Meta AI (FAIR)
`yuandong@meta.com`

## Abstract

We show that Contrastive Learning (CL) under a broad family of loss functions (including InfoNCE) has a unified formulation of coordinate-wise optimization on the network parameter $\boldsymbol{\theta}$ and pairwise importance $\alpha$, where the *max player* $\boldsymbol{\theta}$ learns representation for contrastiveness, and the *min player* $\alpha$ puts more weights on pairs of distinct samples that share similar representations. The resulting formulation, called $\boldsymbol{\alpha}$**-CL**, unifies not only various existing contrastive losses, which differ by how sample-pair importance $\alpha$ is constructed, but also is able to extrapolate to give novel contrastive losses beyond popular ones, opening a new avenue of contrastive loss design. These novel losses yield comparable (or better) performance on CIFAR10, STL-10 and CIFAR-100 than classic InfoNCE. Furthermore, we also analyze the max player in detail: we prove that with fixed $\alpha$, max player is equivalent to Principal Component Analysis (PCA) for deep linear network, and almost all local minima are global and rank-1, recovering optimal PCA solutions. Finally, we extend our analysis on max player to 2-layer ReLU networks, showing that its fixed points can have higher ranks. Codes are available [1].

## 1 Introduction

While contrastive self-supervised learning has been shown to learn good features (Chen et al., 2020; He et al., 2020; Oord et al., 2018) and in many cases, comparable with features learned from supervised learning, it remains an open problem what features it learns, in particular when deep nonlinear networks are used. Theory on this is quite sparse, mostly focusing on loss function (Arora et al., 2019) and treating the networks as a black-box function approximator.

In this paper, we present a novel perspective of contrastive learning (CL) for a broad family of contrastive loss functions $\mathcal{L}(\boldsymbol{\theta})$: minimizing $\mathcal{L}(\boldsymbol{\theta})$ corresponds to a *coordinate-wise optimization* procedure on an objective $\mathcal{E}_\alpha(\boldsymbol{\theta}) - \mathcal{R}(\alpha)$ with respect to network parameter $\boldsymbol{\theta}$ and *pairwise importance* $\alpha$ on batch samples, where $\mathcal{E}_\alpha(\boldsymbol{\theta})$ is an energy function and $\mathcal{R}(\alpha)$ is a regularizer, both associated with the original contrastive loss $\mathcal{L}$. In this view, the *max player $\boldsymbol{\theta}$* learns a representation to maximize the contrastiveness of different samples and keep different augmentation view of the same sample similar, while the *min player $\alpha$* puts more weights on pairs of different samples that appear similar in the representation space, subject to regularization. Empirically, this formulation, named Pair-weighed Contrastive Learning ($\boldsymbol{\alpha}$-CL), when coupled with various regularization terms, yields novel contrastive losses that show comparable (or better) performance in CIFAR10 (Krizhevsky et al., 2009) and STL-10 (Coates et al., 2011).

We then focus on the behavior of the max player who does *representation learning* via maximizing the energy function $\mathcal{E}_\alpha(\boldsymbol{\theta})$. When the underlying network is deep linear, we show that $\max_{\boldsymbol{\theta}} \mathcal{E}_\alpha(\boldsymbol{\theta})$ is the loss function (under re-parameterization) of Principal Component Analysis (PCA) (Wold et al., 1987), a century-old unsupervised dimension reduction method. To further show they are equivalent,

---

[1] `https://github.com/facebookresearch/luckmatters/tree/main/ssl/real-dataset`

36th Conference on Neural Information Processing Systems (NeurIPS 2022).

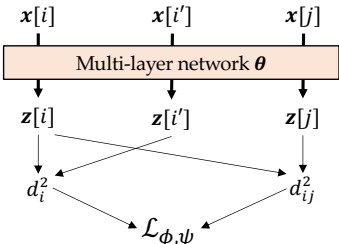

| Contrastive Loss | $\phi(x)$ | $\psi(x)$ |
|---|---|---|
| InfoNCE (Oord et al., 2018) | $\tau\log(\epsilon + x)$ | $e^{x/\tau}$ |
| MINE (Belghazi et al., 2018) | $\log(x)$ | $e^x$ |
| Triplet (Schroff et al., 2015) | $x$ | $[x+\epsilon]_+$ |
| Soft Triplet (Tian et al., 2020c) | $\tau\log(1+x)$ | $e^{x/\tau+\epsilon}$ |
| N+1 Tuplet (Sohn, 2016) | $\log(1+x)$ | $e^x$ |
| Lifted Structured (Oh Song et al., 2016) | $[\log(x)]^2_+$ | $e^{x+\epsilon}$ |
| Modified Triplet Eqn. 10 (Coria et al., 2020) | $x$ | sigmoid$(cx)$ |
| Triplet Contrastive Eqn. 2 (Ji et al., 2021) | linear | linear |

Figure 1: Problem Setting. **Left**: Data points ($i$-th sample $\boldsymbol{x}[i]$ and its augmented version $\boldsymbol{x}[i']$, $j$-th sample $\boldsymbol{x}[j]$) are sent to networks with weights $\boldsymbol{\theta}$, to yield outputs $\boldsymbol{z}[i]$, $\boldsymbol{z}[i']$ and $\boldsymbol{z}[j]$. From the outputs $\boldsymbol{z}$, we compute pairwise squared distance $d_{ij}^2$ between $\boldsymbol{z}[i]$ and $\boldsymbol{z}[j]$ and intra-class squared distance $d_i^2$ between $\boldsymbol{z}[i]$ and $\boldsymbol{z}[i']$ for contrastive learning with a general family of contrastive loss $\mathcal{L}_{\phi,\psi}$ (Eqn. 1). **Right**: Different existing loss functions corresponds to different monotonous functions $\phi$ and $\psi$. Here $[x]_+ := \max(x, 0)$.

we prove that the nonlinear training dynamics of CL with a linear multi-layer feedforward network (MLP) enjoys nice properties: with proper weight normalization, almost all its local optima are global, achieving optimal PCA objective, and are rank-1. The only difference here is that the data augmentation provides negative eigen-directions to avoid.

Furthermore, we extend our analysis to 2-layer ReLU network, to explore the difference between the rank-1 PCA solution and the solution learned by a nonlinear network. Assuming the data follow an orthogonal mixture model, the 2-layer ReLU networks enjoy similar dynamics as the linear one, except for a special *sticky weight rule* that keeps the low-layer weights to be non-negative and stays zero when touching zero. In the case of one hidden node, we prove that the solution in ReLU always picks a single mode from the mixtures. In the case of multiple hidden nodes, the resulting solution is not necessarily rank-1.

## 2 Related Work

**Contrastive learning**. While many contrastive learning techniques (e.g., SimCLR (Chen et al., 2020), MoCo (He et al., 2020), PIRL (Misra & Maaten, 2020), SwAV (Caron et al., 2020), DeepCluster (Caron et al., 2018), Barlow Twins (Zbontar et al., 2021), InstDis (Wu et al., 2018), etc) have been proposed empirically and able to learn good representations for downstream tasks, theoretical study is relatively sparse, mostly focusing on loss function itself (Tian et al., 2020b; HaoChen et al., 2021; Arora et al., 2019), e.g., the relationship of loss functions with mutual information (MI). We are not aware of prior works on joint analysis that combines neural network and loss functions in CL.

**Theoretical analysis of deep networks**. Many works focus on analysis of deep linear networks in supervised setting, where label is given. (Baldi & Hornik, 1989; Zhou & Liang, 2018; Kawaguchi, 2016) analyze the critical points of linear networks. (Saxe et al., 2014; Arora et al., 2018) also analyze the training dynamics. On the other hand, analyzing nonlinear networks has been a difficult task. Existing works mostly lie in supervised learning, e.g., teacher-student setting (Tian, 2020; Allen-Zhu et al., 2018), landscape (Safran & Shamir, 2018). For contrastive learning, recent work (Wen & Li, 2021) analyzes the dynamics of 1-layer ReLU networks with a specific weight structure, and (Jing et al., 2022) analyzes the collapsing behaviors in 2-layer linear network. To our best knowledge, we are unaware of such analysis on networks with $> 2$ layers (linear or nonlinear) in the context of CL.

**Connection between Principal Component Analysis (PCA) and Self-supervised Learning**. (Lee et al., 2021) establishes the statistical connection between non-linear Canonical Component Analysis (CCA) and SimSiam (Chen & He, 2020) for any zero-mean encoder, without considering the aspect of training dynamics. In contrast, we reformulate contrastive learning as coordinate-wise optimization procedure with min/max players, in which the max player is a reparameterization of PCA optimized with gradient descent, and analyze its training dynamics in the presence of specific neural architectures.

## 3 Contrastive Learning as Coordinate-wise Optimization

**Notation**. Suppose we have $N$ pairs of samples $\{\boldsymbol{x}[i]\}_{i=1}^N$ and $\{\boldsymbol{x}[i']\}_{i=1}^N$. Both $\boldsymbol{x}[i]$ and $\boldsymbol{x}[i']$ are augmented samples from sample $i$ and $\boldsymbol{x}$ represents the input batch. These samples are sent to

neural networks and $z[i]$ and $z[i']$ are their outputs. The goal of contrastive learning (CL) is to find the representation to maximize the squared distance $d_{ij}^2 := \|z[i] - z[j]\|_2^2/2$ between distinct samples $i$ and $j$, and minimize the squared distance $d_i^2 := \|z[i] - z[i']\|_2^2/2$ between different data augmentations $x[i]$ and $x[i']$ of the same sample $i$.

## 3.1 A general family of contrastive loss

We consider minimizing a general family of loss functions $\mathcal{L}_{\phi,\psi}$, where $\phi$ and $\psi$ are monotonously increasing and differentiable scalar functions (define $\xi_i := \sum_{j \neq i} \psi(d_i^2 - d_{ij}^2)$ for notation brevity):

$$\min_{\boldsymbol{\theta}} \mathcal{L}_{\phi,\psi}(\boldsymbol{\theta}) := \sum_{i=1}^N \phi(\xi_i) = \sum_{i=1}^N \phi\left(\sum_{j \neq i} \psi(d_i^2 - d_{ij}^2)\right) \tag{1}$$

Both $i$ and $j$ run from 1 to $N$. With different $\phi$ and $\psi$, Eqn. 1 covers many loss functions (Tbl. 1). In particular, setting $\phi(x) = \tau \log(\epsilon + x)$ and $\psi(x) = \exp(x/\tau)$ gives a generalized version of InfoNCE loss (Oord et al., 2018):

$$\mathcal{L}_{nce} := -\tau \sum_{i=1}^N \log \frac{\exp(-d_i^2/\tau)}{\epsilon \exp(-d_i^2/\tau) + \sum_{j \neq i} \exp(-d_{ij}^2/\tau)} = \tau \sum_{i=1}^N \log\left(\epsilon + \sum_{j \neq i} e^{\frac{d_i^2 - d_{ij}^2}{\tau}}\right) \tag{2}$$

where $\epsilon > 0$ is some constant not related to $z[i]$ and $z[i']$. $\epsilon = 1$ has been used in many works (He et al., 2020; Tian et al., 2020a). Setting $\epsilon = 0$ yields SimCLR setting (Chen et al., 2020) where the denominator doesn't contains $\exp(-d_i^2/\tau)$. This is also used in (Yeh et al., 2021).

## 3.2 The other side of gradient descent of contrastive loss

To minimize $\mathcal{L}_{\phi,\psi}$, gradient descent follows its negative gradient direction. As a first discovery of this work, it turns out that the gradient descent of the loss function $\mathcal{L}$ is the *gradient ascent* direction of another energy function $\mathcal{E}_\alpha$:

**Theorem 1.** *For any differential mapping $z = z(x; \boldsymbol{\theta})$, gradient descent of $\mathcal{L}_{\phi,\psi}$ is equivalent to __gradient ascent__ of the objective $\mathcal{E}_\alpha(\boldsymbol{\theta}) := \frac{1}{2}\mathrm{tr}(\mathbb{C}_\alpha[z(\boldsymbol{\theta}), z(\boldsymbol{\theta})])$:*

$$\frac{\partial \mathcal{L}_{\phi,\psi}}{\partial \boldsymbol{\theta}} = -\frac{\partial \mathcal{E}_\alpha}{\partial \boldsymbol{\theta}}\bigg|_{\alpha = \alpha(\boldsymbol{\theta})} \tag{3}$$

*Here the* pairwise importance $\alpha = \alpha(\boldsymbol{\theta}) := \{\alpha_{ij}(\boldsymbol{\theta})\}$ *is a function of input batch $x$, defined as:*

$$\alpha_{ij}(\boldsymbol{\theta}) := \phi'(\xi_i)\psi'(d_i^2 - d_{ij}^2) \geq 0 \tag{4}$$

*where $\phi', \psi' \geq 0$ are derivatives of $\phi, \psi$. The* contrastive covariance $\mathbb{C}_\alpha[\cdot, \cdot]$ *is defined as:*

$$\mathbb{C}_\alpha[\boldsymbol{a}, \boldsymbol{b}] := \sum_{i=1}^N \sum_{j \neq i} \alpha_{ij}(\boldsymbol{a}[i] - \boldsymbol{a}[j])(\boldsymbol{b}[i] - \boldsymbol{b}[j])^\top - \sum_{i=1}^N \left(\sum_{j \neq i} \alpha_{ij}\right)(\boldsymbol{a}[i] - \boldsymbol{a}[i'])(\boldsymbol{b}[i] - \boldsymbol{b}[i'])^\top \tag{5}$$

*That is, __minimizing__ the loss function $\mathcal{L}_{\phi,\psi}(\boldsymbol{\theta})$ can be regarded as __maximizing__ the energy function $\mathcal{E}_{\alpha = \mathrm{sg}(\alpha(\boldsymbol{\theta}))}(\boldsymbol{\theta})$ with respect to $\boldsymbol{\theta}$. Here $\mathrm{sg}(\cdot)$ means stop-gradient, i.e., the gradient of $\boldsymbol{\theta}$ is not backpropagated into $\alpha(\boldsymbol{\theta})$.*

Please check Supplementary Materials (SM) for all proofs. From the definition of energy $\mathcal{E}_\alpha(\boldsymbol{\theta})$, it is clear that $\alpha_{ij}$ determines the importance of each sample pair $x[i]$ and $x[j]$. For $(i, j)$-pair that "deserves attention", $\alpha_{ij}$ is large so that it plays a large role in the contrastive covariance term. In particular, for InfoNCE loss with $\epsilon = 0$, the pairwise importance $\alpha$ takes the following form:

$$\alpha_{ij} = \frac{\exp(-d_{ij}^2/\tau)}{\sum_{j \neq i} \exp(-d_{ij}^2/\tau)} > 0 \tag{6}$$

which means that InfoNCE focuses on $(i, j)$-pair with small squared distance $d_{ij}^2$. If both $\phi$ and $\psi$ are linear, then $\alpha_{ij} = \mathrm{const}$ and $\mathcal{L}$ is a simple subtraction of positive/negative squared distances.

From Thm. 1, an important observation is that when propagating gradient w.r.t. $\boldsymbol{\theta}$ using the objective $\mathcal{E}_\alpha(\boldsymbol{\theta})$ during the backward pass, the gradient does not propagate into $\alpha(\boldsymbol{\theta})$, even if $\alpha(\boldsymbol{\theta})$ is a function of $\boldsymbol{\theta}$ in the forward pass. In fact, in Sec. 6 we show that propagating gradient through $\alpha(\boldsymbol{\theta})$ yields worse empirical performance. This suggests that $\alpha$ should be treated as an *independent* variable when optimizing $\boldsymbol{\theta}$. It turns out that if $\psi(x)$ is an exponential function (as in most cases of Tbl. 1), this is indeed true and $\alpha$ can be determined by a separate optimization procedure:

**Theorem 2.** *If $\psi(x) = e^{x/\tau}$, then the corresponding pairwise importance $\alpha$ (Eqn. 4) is the solution to the minimization problem:*

$$\alpha(\boldsymbol{\theta}) = \arg\min_{\alpha \in \mathcal{A}} \mathcal{E}_\alpha(\boldsymbol{\theta}) - \mathcal{R}(\alpha), \qquad \mathcal{A} := \left\{ \alpha : \quad \forall i, \ \sum_{j \neq i} \alpha_{ij} = \tau^{-1} \xi_i \phi'(\xi_i), \ \alpha_{ij} \geq 0 \right\} \quad (7)$$

*Here the regularization $\mathcal{R}(\alpha) = \mathcal{R}_H(\alpha) := \tau \sum_{i=1}^N H(\alpha_{i\cdot}) = -\tau \sum_{i=1}^N \sum_{j \neq i} \alpha_{ij} \log \alpha_{ij}$.*

For InfoNCE, the feasible set $\mathcal{A}$ becomes $\{\alpha : \alpha \geq 0, \sum_{j \neq i} \alpha_{ij} = \xi_i/(\xi_i + \epsilon)\}$. This means that if $i$-th sample is already well-separated (small intra-augmentation distance $d_i$ and large inter-augmentation distance $d_{ij}$), then $\xi_i$ is small, the summation of weights $\sum_{j \neq i} \alpha_{ij}$ associated with sample $i$ is also small and such a sample is overall discounted. Setting $\epsilon = 0$ reduces to sample-agnostic constraint (i.e., $\sum_{j \neq i} \alpha_{ij} = 1$).

Thm. 2 leads to a novel perspective of *coordinate-wise optimization* for Contrastive Learning (CL):

**Corollary 1** (Contrastive Learning as Coordinate-wise Optimization)**.** *If $\psi(x) = e^{x/\tau}$, minimizing $\mathcal{L}_{\phi,\psi}$ is equivalent to the following iterative procedure:*

$$\textit{(Min-player } \alpha) \qquad \qquad \alpha_t = \arg\min_{\alpha \in \mathcal{A}} \mathcal{E}_\alpha(\boldsymbol{\theta}_t) - \mathcal{R}(\alpha) \qquad \qquad (8a)$$

$$\textit{(Max-player } \boldsymbol{\theta}) \qquad \qquad \boldsymbol{\theta}_{t+1} = \boldsymbol{\theta}_t + \eta \nabla_{\boldsymbol{\theta}} \mathcal{E}_{\alpha_t}(\boldsymbol{\theta}) \qquad \qquad (8b)$$

Intuitively, the max player $\boldsymbol{\theta}$ (Eqn. 8b) performs one-step gradient ascent for the objective $\mathcal{E}_\alpha(\boldsymbol{\theta}) - \mathcal{R}(\alpha)$, *learns a representation* to maximize the distance of different samples and minimize the distance of the same sample with different augmentations (as suggested by $\mathbb{C}_\alpha[\boldsymbol{z}, \boldsymbol{z}]$). On the other hand, the "min player" $\alpha$ (Eqn. 8a) finds optimal $\alpha$ analytically, assigning high weights on confusing pairs for "max player" to solve.

**Relation to max-min formulation**. While Corollary 1 looks very similar to max-min formulation, important differences exist. Different from traditional max-min formulation, in Corollary 1 there is asymmetry between $\boldsymbol{\theta}$ and $\alpha$. First, $\boldsymbol{\theta}$ only follows one step update along gradient ascent direction of $\max_{\boldsymbol{\theta}} \mathcal{E}_\alpha(\boldsymbol{\theta})$, while $\alpha$ is solved analytically. Second, due to the stop-gradient operator, the gradient of $\boldsymbol{\theta}$ contains no knowledge on how $\boldsymbol{\theta}$ changes $\alpha$. This prevents $\boldsymbol{\theta}$ from adapting to $\alpha$'s response on changing $\boldsymbol{\theta}$. Both give advantages to min-player $\alpha$ to find the confusing sample pairs more effectively.

**Relation to hard-negative samples**. While many previous works (Kalantidis et al., 2020; Robinson et al., 2021) focus on seeking and putting more weights on hard samples, Corollary 1 shows that contrastive losses already have such mechanism at the batch level, focusing on "hard-negative pairs" beyond hard-negative samples.

From this formulation, different pairwise importance $\alpha$ corresponds to different loss functions within the loss family specified by Eqn. 1, and choosing among this family (i.e., different $\phi$ and $\psi$) can be regarded as choosing different $\alpha$ when optimizing the *same* objective $\mathcal{E}_\alpha(\boldsymbol{\theta})$. Based on this observation, we now propose the following training framework called $\boldsymbol{\alpha}$**-CL**:

**Definition 1** (P̲air-weighed C̲ontrastive L̲earning ($\alpha$-CL))**.** *Optimize $\boldsymbol{\theta}$ by gradient ascent: $\boldsymbol{\theta}_{t+1} = \boldsymbol{\theta}_t + \eta \nabla_{\boldsymbol{\theta}} \mathcal{E}_{\mathrm{sg}(\alpha_t)}(\boldsymbol{\theta})$, with the energy $\mathcal{E}_\alpha(\boldsymbol{\theta})$ defined in Thm. 1 and pairwise importance $\alpha_t = \alpha(\boldsymbol{\theta}_t)$.*

In $\alpha$-CL, choosing $\alpha$ can be achieved by either implicitly specifying a regularizer $\mathcal{R}(\alpha)$ and solve Eqn. 8a, or by a direct mapping $\alpha = \alpha(\boldsymbol{\theta})$ without any optimization. This opens a novel revenue for CL loss design. Initial experiments (Sec. 6) show that $\alpha$-CL gives comparable (or even better) downstream performance in CIFAR10 and STL-10, compared to vanilla InfoNCE loss.

## 4 Representation Learning in Deep Linear CL is PCA

In Corollary 1, optimizing over $\alpha$ is well-understood, since $\mathcal{E}_\alpha(\boldsymbol{\theta})$ is *linear* w.r.t. $\alpha$ and $\mathcal{R}(\alpha)$ in general is a (strong) concave function. As a result, $\alpha$ has a unique optimal. On the other hand,

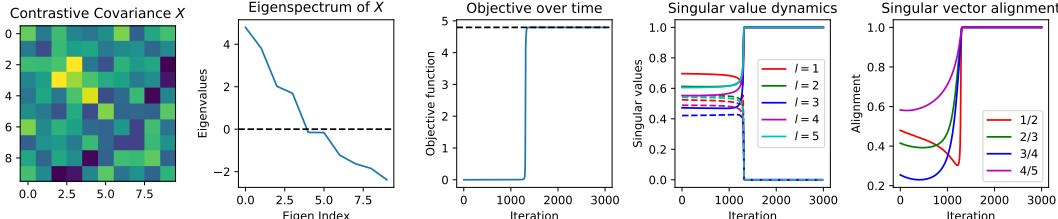

Figure 2: Dynamics of CL with multilayer ($L = 5$) linear network (`DeepLin`) with fixed $\alpha$. Running the training dynamics (Lemma 1) quickly leads to convergence towards the maximal eigenvalue of $X_\alpha$. For dynamics of singular value of $W_l$, the largest singular values (solid lines) converges to 1 while the second largest singular values (dashed lines) decay to 0.

understanding the max player $\max_{\boldsymbol{\theta}} \mathcal{E}_\alpha(\boldsymbol{\theta})$ is important since it performs *representation learning* in CL. It is also a hard problem because of non-convex optimization.

We start with a specific case when $\boldsymbol{z}$ is a deep linear network, i.e., $\boldsymbol{z} = W(\boldsymbol{\theta})\boldsymbol{x}$, where $W$ is the equivalent linear mapping for the deep linear network, and $\boldsymbol{\theta}$ is the parameters to be optimized. Note that this covers many different kinds of deep linear networks, including VGG-like (Saxe et al., 2014), ResNet-like (Hardt & Ma, 2017) and DenseNet-like (Huang et al., 2017). For notation brevity, we define $\mathbb{C}_\alpha[\boldsymbol{x}] := \mathbb{C}_\alpha[\boldsymbol{x}, \boldsymbol{x}]$.

**Corollary 2** (Representation learning in Deep Linear CL reparameterizes Principal Component Analysis (PCA)). *When $\boldsymbol{z} = W(\boldsymbol{\theta})\boldsymbol{x}$ with a constraint $WW^\top = I$, $\mathcal{E}_\alpha$ is the objective of Principal Component Analysis (PCA) with reparameterization $W = W(\boldsymbol{\theta})$:*

$$\max_{\boldsymbol{\theta}} \mathcal{E}_\alpha(\boldsymbol{\theta}) = \frac{1}{2}\mathrm{tr}(W(\boldsymbol{\theta})X_\alpha W^\top(\boldsymbol{\theta})) \quad \text{s.t. } WW^\top = I \tag{9}$$

*here $X_\alpha := \mathbb{C}_\alpha[\boldsymbol{x}]$ is the contrastive covariance of input $\boldsymbol{x}$.*

As a comparison, in traditional Principal Component Analysis, the objective is (Kokiopoulou et al., 2011): $\frac{1}{2}\max_W \mathrm{tr}(W\mathbb{V}_{\mathrm{sample}}[\boldsymbol{x}]W^\top)$ subject to the constraint $WW^\top = I$, where $\mathbb{V}_{\mathrm{sample}}[\boldsymbol{x}]$ is the empirical covariance of the dataset (here it is one batch). Therefore, $X_\alpha$ can be regarded as a generalized covariance matrix, possibly containing negative eigenvalues. In the case of supervised CL (i.e,. pairs from the same/different labels are treated as positive/negative (Khosla et al., 2020)), then it is connected with Fisher's Linear Discriminant Analysis (Fisher, 1936).

Here we show a mathematically rigorous connection between CL and dimensional reduction, as suggested intuitively in (Hadsell et al., 2006). Unlike traditional PCA, due to the presence of data augmentation, while symmetric, the contrastive covariance $X_\alpha$ is not necessarily a PSD matrix. Nevertheless, the intuition is the same: to find the direction that corresponds to maximal variation of the data.

While it is interesting to discover that CL with deep linear network is essentially a reparameterization of PCA, it remains elusive that such a reparameterization leads to the same solution of PCA, in particular when the network is deep (and may contain local optima). Also, PCA has an overall end-to-end constraint $WW^\top = I$, while in network training, we instead use normalization layers and it is unclear whether they are equivalent or not.

In this section, we show for a specific deep linear model, almost all its local maxima of Eqn. 9 are global and it indeed solves PCA.

### 4.1 A concrete deep linear model

We study a concrete deep linear network with parameters/weights $\boldsymbol{\theta} := \{W_l\}_{l=1}^L$:

$$\boldsymbol{z}[i] := W_L W_{L-1} \ldots W_1 \boldsymbol{x}[i] \tag{10}$$

Here $W_l \in \mathbb{R}^{n_l \times n_{l-1}}$, $n_l$ is the number of nodes at layer $l$, $\boldsymbol{z}[i]$ is the output of $\boldsymbol{x}[i]$ and similarly $\boldsymbol{z}[i']$ for $\boldsymbol{x}[i']$. We use $\boldsymbol{\theta}$ to represent the collection of weights at all layers. For convenience, we define the $l$-th layer activation $\boldsymbol{f}_l[i] = W_l \boldsymbol{f}_{l-1}[i]$. With this notation $\boldsymbol{f}_0[i] = \boldsymbol{x}[i]$ is the input and $\boldsymbol{z}[i] = W_L \boldsymbol{f}_{L-1}[i]$.

We call this setting `DeepLin`. The Jacobian matrix $W_{>l} := W_L W_{L-1} \dots W_{l+1}$ and $W := W_{>0} = W_L W_{L-1} \dots W_1$.

**Lemma 1.** *The training dynamics in* `DeepLin` *is* $\dot{W}_l = W_{>l}^\top W_{>l} W_l \mathbb{C}_\alpha[\boldsymbol{f}_{l-1}]$

Note that $\mathbb{C}_\alpha[\boldsymbol{f}_0] = \mathbb{C}_\alpha[\boldsymbol{x}] = X_\alpha$. Similar to supervised learning (Arora et al., 2018; Du et al., 2018b), nearby layers are also balanced: $\frac{\mathrm{d}}{\mathrm{d}t} \left( W_l W_l^\top - W_{l+1}^\top W_{l+1} \right) = 0$.

## 4.2 Normalization Constraints

Note that if we just run the training dynamics (Lemma 1) without any constraints, $\|W_l\|_F$ will go to infinity. Fortunately, empirical works already suggest various ways of normalization to stabilize the network training.

One popular technique in CL is $\ell_2$ normalization. It is often put right after the output of the network and before the loss function $\mathcal{L}$ (Chen et al., 2020; Grill et al., 2020; He et al., 2020), i.e., $\hat{\boldsymbol{z}}[i] = \boldsymbol{z}[i]/\|\boldsymbol{z}[i]\|_2$. Besides, LayerNorm (Ba et al., 2016) (i.e., $\hat{\boldsymbol{f}}[i] = (\boldsymbol{f}[i] - \mathrm{mean}(\boldsymbol{f}[i]))/\mathrm{std}(\boldsymbol{f}[i])$) is extensively used in Transformer-based models (Xiong et al., 2020). Here we show that for gradient flow dynamics of MLP models, such normalization layers conserve $\|W_l\|_F$ for any $l$ below it, regardless of loss function.

**Lemma 2.** *For MLP, if the weight $W_l$ is below a $\ell_2$-norm or LayerNorm layer, then $\frac{\mathrm{d}}{\mathrm{d}t} \|W_l\|_F^2 = 0$.*

Note that Lemma 2 also holds for nonlinear MLP with reversible activations, which includes ReLU (see SM). Therefore, without loss of generality, we consider the following complete objective for max player with `DeepLin` (here $\Theta$ is the constraint set of the weights due to normalization):

$$\max_{\boldsymbol{\theta} \in \Theta} \mathcal{E}_\alpha(\boldsymbol{\theta}) := \frac{1}{2}\mathrm{tr}(W X_\alpha W^\top), \quad \Theta := \{\boldsymbol{\theta} : \|W_l\|_F = 1, 1 \le l \le L\} \tag{11}$$

## 4.3 Representation Learning with `DeepLin` is PCA

As one of our main contributions, the following theorem asserts that almost all local optimal solutions of Eqn. 11 are global, and the optimal objective corresponds to the PCA objective. Note that (Kawaguchi, 2016; Laurent & Brecht, 2018) proves no bad local optima for deep linear network in supervised learning, while here we give similar results for CL, and additionally we also give the (simple) rank-1 structure of all local optima.

**Theorem 3** (Representation Learning with `DeepLin` is PCA). *If $\lambda_{\max}(X_\alpha) > 0$, then for any local maximum $\boldsymbol{\theta} \in \Theta$ of Eqn. 11 whose $W_{>1}^\top W_{>1}$ has distinct maximal eigenvalue:*

- *there exists a set of unit vectors $\{\boldsymbol{v}_l\}_{l=0}^L$ so that $W_l = \boldsymbol{v}_l \boldsymbol{v}_{l-1}^\top$ for $1 \le l \le L$, in particular, $\boldsymbol{v}_0$ is the unit eigenvector corresponding to $\lambda_{\max}(X_\alpha)$,*

- *$\boldsymbol{\theta}$ is global optimal with objective $2\mathcal{E}^* = \lambda_{\max}(X_\alpha)$.*

**Corollary 3.** *If we additionally use per-filter normalization (i.e., $\|\boldsymbol{w}_{lk}\|_2 = 1/\sqrt{n_l}$), then Thm. 3 holds and $\boldsymbol{v}_l$ is more constrained: $[\boldsymbol{v}_l]_k = \pm 1/\sqrt{n_l}$ for $1 \le l \le L - 1$.*

**Remark**. Here we prove that given fixed $\alpha$, maximizing $\mathcal{E}_\alpha(\boldsymbol{\theta})$ gives rank-1 solutions for deep linear network. This conclusion is an extension of (Jing et al., 2022), which shows weight collapsing happens if $\boldsymbol{\theta}$ is 2-layer linear network and $\alpha$ is fixed. If the pairwise importance $\alpha$ is adversarial, then it may not lead to a rank-1 solution. In fact, $\alpha$ can magnify minimal eigen-directions and change the eigenstructure of $X_\alpha$ continuously. We leave it for future work.

Note that the condition that "$W_{>1}^\top W_{>1}$ has distinct maximal eigenvalue" is important. Otherwise there are counterexamples. For example, consider 1-layer linear network $\boldsymbol{z} = W_1 \boldsymbol{x}$, and $X_\alpha$ has duplicated maximal eigenvalues (with $\boldsymbol{u}_1$ and $\boldsymbol{u}_2$ being corresponding orthogonal eigenvectors), then $W_{>1}^\top W_{>1} = I$ (i.e., it has degenerated eigenvalues), and for any local maximal $W_1$, its row vector can be arbitrary linear combinations of $\boldsymbol{u}_1$ and $\boldsymbol{u}_2$ and thus $W_1$ is not rank-1.

Compared to recent works (Ji et al., 2021) that also relates CL with PCA in linear representation setting using constant $\alpha$, our Theorem 3 has no statistical assumptions on data distribution and augmentation, and operates on vanilla InfoNCE loss and deep architectures.

# 5 How Representation Learning Differs in Two-layer ReLU Network

So far we have shown that the max player $\max_{\boldsymbol{\theta}} \mathcal{E}_\alpha(\boldsymbol{\theta}) := \frac{1}{2}\mathrm{tr}(\mathbb{C}_\alpha[\boldsymbol{z}(\boldsymbol{\theta})])$ is essentially a PCA objective when the input-output mapping $\boldsymbol{z} = W(\boldsymbol{\theta})\boldsymbol{x}$ is linear. A natural question arises. What is the benefit of CL if its representation learning component has such a simple nature? Why can it learn a good representation in practice beyond PCA?

For this, nonlinearity is the key but understanding its role is highly nontrivial. For example, when the neural network model is nonlinear, Thm. 1 and Corollary 1 holds but *not* Corollary 2. Therefore, there is not even a well-defined $X_\alpha$ due to the fact that multiple hidden nodes can be switched on/off given different data input. Previous works (Safran & Shamir, 2018; Du et al., 2018a) also show that with nonlinearity, in supervised learning spurious local optima exist.

Here we take a first step to analyze nonlinear cases. We study 2-layer models with ReLU activation $h(x) = \max(x, 0)$. We show that with a proper data assumption, the 2-layer model shares a *modified* version of dynamics with its linear version, and the contrastive covariance term $X_\alpha$ (and its eigenstructure) remains well-defined and useful in nonlinear case.

## 5.1 The 2-layer ReLU network and data model

We consider the bottom-layer weight $W_1 = [\boldsymbol{w}_{11}, \boldsymbol{w}_{12}, \ldots, \boldsymbol{w}_{1K}]^\top$ with $\boldsymbol{w}_{1k}$ being the $k$-th filter. For brevity, let $K = n_1$ be the number of hidden nodes. We still consider solution in the constraint set $\Theta$ (Eqn. 11), since Lemma 2 still holds for ReLU networks. This model is named `ReLU2Layer`.

In addition, we assume the following data model:

**Assumption 1** (Orthogonal mixture model within receptive field $R_k$). *There exists a set of orthonormal bases $\{\bar{\boldsymbol{x}}_m\}_{m=1}^M$ so that any input data $\boldsymbol{x}[i] = \sum_m a_m[i]\bar{\boldsymbol{x}}_m$ satisfies the property that $a_m[i]$ is **Nonnegative**: $a_m[i] \geq 0$, **One-hot**: for any $k$, $a_m[i] > 0$ for at most one $m$ and **Augmentation** only scales $\boldsymbol{x}_k$ by a (sample-dependent) factor, i.e., $\boldsymbol{x}[i'] = \gamma[i]\boldsymbol{x}[i]$ with $\gamma[i] > 0$.*

Since all $\boldsymbol{x}$ appears in the inner-product with the weight vectors $\boldsymbol{w}_{1k}$, with a rotation of coordination, we can just set $\bar{\boldsymbol{x}}_m = \boldsymbol{e}_m$, where $\boldsymbol{e}_m$ is the one-hot vector with $m$-th component being 1. In this case, $\boldsymbol{x} \geq 0$ is always a one-hot vector with only at most only one positive entry.

Intuitively, the model is motivated by sparsity: in each instantiation of $\boldsymbol{x}$, there are very small number of activated modes and their linear combination becomes the input signal $\boldsymbol{x}$. As we shall see, even with this simple model, the dynamics of ReLU network behaves very differently from the linear case.

With this assumption, we only need to consider nonnegative low-layer weights and $X_\alpha$ is still a valid quantity for `ReLU2Layer`:

**Lemma 3** (Evaluation of `ReLU2Layer`). *If Assumption 1 holds, setting $\boldsymbol{w}'_{1k} = \max(\boldsymbol{w}_{1k}, 0)$ won't change the output of `ReLU2Layer`. Furthermore, if $W_1 \geq 0$, then the formula for linear network $\mathcal{E}_\alpha = \frac{1}{2}\mathrm{tr}(W_2 W_1 X_\alpha W_1^\top W_2^\top)$ still works for `ReLU2Layer`.*

On the other hand, sharing the energy function $\mathcal{E}_\alpha$ does not mean `ReLU2Layer` is completely identical to its linear version. In fact, the dynamics follows its linear counterparts, but with important modifications:

**Theorem 4** (Dynamics of `ReLU2Layer`). *If Assumption 1 holds, then the dynamics of `ReLU2Layer` with $\boldsymbol{w}_{1k} \geq 0$ is equivalent to linear dynamics with the **Sticky Weight rule**: any component that reaches 0 stays 0.*

As we will see, this modification leads to very different dynamics and local optima in `ReLU2Layer` from linear cases, even when there is only one ReLU node.

## 5.2 Dynamics in One ReLU node

Now we consider the dynamics of the simplest case: `ReLU2Layer` with only 1 hidden node. In this case, $W_{>1}^\top W_{>1}$ is a scalar and thus $W_2^\top W_2 = \mathrm{tr}(W_2^\top W_2) = 1$. We only need to consider $\boldsymbol{w}_1 \in \mathbb{R}^{n_1}$, which is the only weight vector in the lower layer, under the constraint $\|W_1\|_F = \|\boldsymbol{w}_1\|_2 = 1$ (Eqn. 11). We denote this setting as `ReLU2Layer1Hid`.

The dynamics now becomes very different from linear setting. Under linear network, according to Theorem 3, $\boldsymbol{w}_1$ converges to the largest eigenvector of $X_\alpha = \mathbb{C}_\alpha[\boldsymbol{x}_1]$. For `ReLU2Layer1Hid`, situation differs drastically:

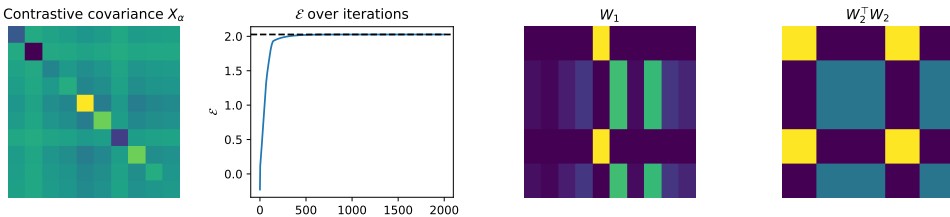

Figure 3: Theorem 6 shows that training `ReLU2Layer` could lead to more diverse hidden weight patterns beyond rank-1 solution obtained in the linear case (shown in right two figures: converged $W_1$ and $W_2^\top W_2$).

**Theorem 5.** *If Assumption 1 holds, then in* `ReLU2Layer1Hid`*, $w_1 \to e_m$ for certain $m$.*

Intuitively, this theorem is achieved by closely tracing the dynamics. When the number of positive entries of $w_1$ is more than 1, the linear dynamics always hits the boundary of the polytope $w_1 \geq 0$, making one of its entry be zero, and stick to zero due to sticky weight rule. This procedure repeats until there is only one survival positive entry in $w_1$.

Overall, this simple case already shows that nonlinear landscape can lead to many local optima: for any $m$, $w_1 = e_m$ is one local optimal. Which one the training falls into depends on weight initialization, and critically affects the properties of per-trained models.

### 5.3 Multiple hidden nodes

For complicated situations like multiple hidden units, completely characterizing the training dynamics like Theorem 5 becomes hard (if not impossible). Instead, we focus on fixed point analysis.

For deep linear model, using multiple hidden units does not lead to any better solutions. According to Thm. 3, at local optimal, $W_1 = v_1 v_0^\top$. This means that the weights $w_{1k}$, which are row vectors of $W_1$, are just a scaled version of the maximal eigenvector $v_0$ of $X_\alpha$. Moreover, this is independent of the eigenstructure of $X_\alpha$ as long as $\lambda_{\max}(X_\alpha) > 0$.

In `ReLU2Layer`, the situation is a bit different. Thm. 6 shows that these hidden nodes are (slightly) more diverse. Fig. 3 shows one such example. The intuition here is that in nonlinear case, rank-1 structure of the critical points may be replaced with low-rank structures.

**Theorem 6** (`ReLU2Layer` encourages diversity)**.** *If Assumption 1 holds, then for any local optimal $(W_2, W_1) \in \Theta$ of* `ReLU2Layer` *with $\mathcal{E} > 0$, either $W_1 = v e_m^\top$ for some $m$ and $v \geq 0$, or* $\mathrm{rank}(W_1) > 1$.

## 6 Experiments

We evaluate our $\alpha$-CL framework (Def. 1) in CIFAR10 (Krizhevsky et al., 2009) and STL-10 (Coates et al., 2011) with ResNet18 (He et al., 2016), and compare the downstream performance of multiple losses, with regularizers taking the form of $\mathcal{R}(\alpha) = \sum_i \sum_{j \neq i} r(\alpha_{ij})$ with a constraint $\sum_{j \neq i} \alpha_{ij} = 1$. Here $r$ can be different concave functions:

- ($\alpha$-CL-$r_H$) Entropy regularizer $r_H(\alpha_{ij}) = -\tau \alpha_{ij} \log \alpha_{ij}$;
- ($\alpha$-CL-$r_\gamma$) Inverse regularizers $r_\gamma(\alpha_{ij}) = \frac{\tau}{1-\gamma} \alpha_{ij}^{1-\gamma}$ ($\gamma > 1$).
- ($\alpha$-CL-$r_s$) Square regularizer $r_s(\alpha_{ij}) = -\frac{\tau}{2} \alpha_{ij}^2$.

Besides, we also compare with the following:

- Minimizing InfoNCE or quadratic loss: $\min_\theta \mathcal{L}(\theta)$ for $\mathcal{L} \in \{\mathcal{L}_{nce}, \mathcal{L}_{quadratic}\}$.
- Setting $\alpha$ as InfoNCE (Eqn. 6) and backpropagates through $\alpha = \alpha(\theta)$ with respect to $\theta$.
- ($\alpha$-CL-direct) Directly setting $\alpha$ (here $p > 1$):

$$\alpha_{ij} = \frac{\exp(-d_{ij}^p/\tau)}{\sum_j \exp(-d_{ij}^p/\tau)} \tag{12}$$

|  | CIFAR-10 | | | STL-10 | | |
|---|---|---|---|---|---|---|
|  | 100 epochs | 300 epochs | 500 epochs | 100 epochs | 300 epochs | 500 epochs |
| $\mathcal{L}_{quadratic}$ | $63.59 \pm 2.53$ | $73.02 \pm 0.80$ | $73.58 \pm 0.82$ | $55.59 \pm 4.00$ | $64.97 \pm 1.45$ | $67.28 \pm 1.21$ |
| $\mathcal{L}_{nce}$ | $84.06 \pm 0.30$ | $87.63 \pm 0.13$ | $87.86 \pm 0.12$ | $78.46 \pm 0.24$ | $82.49 \pm 0.26$ | $83.70 \pm 0.12$ |
| backprop $\alpha(\boldsymbol{\theta})$ | $83.42 \pm 0.25$ | $87.18 \pm 0.19$ | $87.48 \pm 0.21$ | $77.88 \pm 0.17$ | $81.86 \pm 0.30$ | $83.19 \pm 0.16$ |
| $\alpha$-CL-$r_H$ | $84.27 \pm 0.24$ | $87.75 \pm 0.25$ | $87.92 \pm 0.24$ | $78.53 \pm 0.35$ | $82.62 \pm 0.15$ | $83.74 \pm 0.18$ |
| $\alpha$-CL-$r_\gamma$ | $83.72 \pm 0.19$ | $87.51 \pm 0.11$ | $87.69 \pm 0.09$ | $78.22 \pm 0.28$ | $82.19 \pm 0.52$ | $83.47 \pm 0.34$ |
| $\alpha$-CL-$r_s$ | $84.72 \pm 0.10$ | $86.62 \pm 0.17$ | $86.74 \pm 0.15$ | $76.95 \pm 1.06$ | $80.64 \pm 0.77$ | $81.65 \pm 0.59$ |
| $\alpha$-CL-direct | $\mathbf{85.11 \pm 0.19}$ | $\mathbf{87.93 \pm 0.16}$ | $\mathbf{88.09 \pm 0.13}$ | $\mathbf{79.32 \pm 0.36}$ | $\mathbf{82.95 \pm 0.17}$ | $\mathbf{84.05 \pm 0.20}$ |

Table 1: Comparison over multiple loss formulations (ResNet18 backbone, batchsize 128). Top-1 accuracy with linear evaluation protocol. Temperature $\tau = 0.5$ and learning rate is 0.01. **Bold** is highest performance and blue is second highest. Each setting is repeated 5 times with different random seeds.

|  | ResNet18 Backbone | | | ResNet50 Backbone | | |
|---|---|---|---|---|---|---|
|  | CIFAR-100 | | | | | |
|  | 100 epochs | 300 epochs | 500 epochs | 100 epochs | 300 epochs | 500 epochs |
| $\mathcal{L}_{nce}$ | $55.70 \pm 0.37$ | $59.71 \pm 0.36$ | $59.89 \pm 0.34$ | $60.16 \pm 0.48$ | $65.40 \pm 0.31$ | $65.53 \pm 0.30$ |
| $\alpha$-CL-direct | $\mathbf{57.63 \pm 0.07}$ | $\mathbf{60.12 \pm 0.26}$ | $\mathbf{60.27 \pm 0.29}$ | $\mathbf{62.93 \pm 0.28}$ | $\mathbf{65.84 \pm 0.14}$ | $\mathbf{65.87 \pm 0.21}$ |
|  | CIFAR-10 | | | | | |
| $\mathcal{L}_{nce}$ | $84.06 \pm 0.30$ | $87.63 \pm 0.13$ | $87.86 \pm 0.12$ | $86.39 \pm 0.16$ | $89.97 \pm 0.14$ | $90.19 \pm 0.23$ |
| $\alpha$-CL-direct | $\mathbf{85.11 \pm 0.19}$ | $\mathbf{87.93 \pm 0.16}$ | $\mathbf{88.09 \pm 0.13}$ | $\mathbf{87.79 \pm 0.25}$ | $\mathbf{90.41 \pm 0.18}$ | $\mathbf{90.50 \pm 0.21}$ |
|  | STL-10 | | | | | |
| $\mathcal{L}_{nce}$ | $78.46 \pm 0.24$ | $82.49 \pm 0.26$ | $83.70 \pm 0.12$ | $81.64 \pm 0.24$ | $86.57 \pm 0.17$ | $\mathbf{87.90 \pm 0.22}$ |
| $\alpha$-CL-direct | $\mathbf{79.32 \pm 0.36}$ | $\mathbf{82.95 \pm 0.17}$ | $\mathbf{84.05 \pm 0.20}$ | $\mathbf{83.20 \pm 0.25}$ | $\mathbf{87.17 \pm 0.14}$ | $87.85 \pm 0.21$ |

Table 2: More experiments with ResNet18/ResNet50 backbone on CIFAR-10, STL-10 and CIFAR-100. Batchsize is 128. For ResNet18, learning rate is 0.01; for ResNet50, learning rate is 0.001.

For inverse regularizer $r_\gamma$, we pick $\gamma = 2$ and $\tau = 0.5$; for direct-set $\alpha$, we pick $p = 4$ and $\tau = 0.5$; for square regularizer, we use $\tau = 5$. All training is performed with Adam (Kingma & Ba, 2014) optimizer. Code is written in PyTorch and a single modern GPU suffices for the experiments.

The results are shown in Tbl. 1. We can see that (1) backpropagating through $\alpha(\boldsymbol{\theta})$ is worse, justifying our perspective of coordinate-wise optimization, (2) our proposed $\alpha$-CL works for different regularizers, (3) using different regularizer leads to comparable or better performance than original InfoNCE $\mathcal{L}_{nce}$, (4) the pairwise importance $\alpha$ does not even need to come from a minimization process. Instead, we can directly set $\alpha$ based on pairwise squared distances $d_{ij}^2$ and $d_i^2$. For $\alpha$-CL-direct, the performance is slightly worse if we do not normalize $\alpha_{ij}$ (i.e., $\alpha_{ij} := \exp(-d_{ij}^p/\tau)$). It seems that for strong performance, $\frac{\mathrm{d}r}{\mathrm{d}\alpha_{ij}}$ should go to $+\infty$ when $\alpha_{ij} \to 0$. Regularizers that do not satisfy this condition (e.g., squared regularizer $r_s$) may not work as well.

Tbl. 2 shows more experiments with different backbones (e.g., ResNet50) and more complicated datasets (e.g., CIFAR-100). Overall, we see consistent gains of $\alpha$-CL over InfoNCE in early stages of the training (e.g., 1-2 point of absolute percentage gain) and comparable performance at 500 epoch. More ablations on batchsizes and exponent $p$ in Eqn. 12 are provided in Appendix **??**.

## 7 Conclusion and Future Work

We provide a novel perspective of contrastive learning (CL) via the lens of coordinate-wise optimization and propose a unified framework called $\alpha$-CL that not only covers a broad family of loss functions including InfoNCE, but also allows a direct set of importance of sample pairs. Preliminary experiments on CIFAR10/STL-10/CIFAR100 show comparable/better performance with the new loss than InfoNCE. Furthermore, we prove that with deep linear networks, the representation learning part is equivalent to Principal Component Analysis (PCA). In addition, we also extend our analysis to representation learning in 2-layer ReLU network, shedding light on the important difference in representation learning for linear/nonlinear cases.

**Future work**. Our framework $\alpha$-CL turns various loss functions into a unified framework with different choices of pairwise importance $\alpha$ and how to find good choices remains open. Also, we mainly focus on representation learning with fixed pairwise importance $\alpha$. However, in the actual training, $\alpha$ and $\boldsymbol{\theta}$ change concurrently. Understanding their interactions is an important next step. Finally, removing Assumption 1 in ReLU analysis is also an open problem to be addressed later.

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
