# A Proofs

## A.1 Section 3

**Theorem 1.** *For any differential mapping $\boldsymbol{z} = \boldsymbol{z}(\boldsymbol{x}; \boldsymbol{\theta})$, gradient descent of $\mathcal{L}_{\phi,\psi}$ is equivalent to* **_gradient ascent_** *of the objective $\mathcal{E}_\alpha(\boldsymbol{\theta}) := \frac{1}{2}\mathrm{tr}(\mathbb{C}_\alpha[\boldsymbol{z}(\boldsymbol{\theta}), \boldsymbol{z}(\boldsymbol{\theta})])$:*

$$\frac{\partial \mathcal{L}_{\phi,\psi}}{\partial \boldsymbol{\theta}} = -\frac{\partial \mathcal{E}_\alpha}{\partial \boldsymbol{\theta}}\bigg|_{\alpha=\alpha(\boldsymbol{\theta})} \tag{3}$$

*Here the pairwise importance $\alpha = \alpha(\boldsymbol{\theta}) := \{\alpha_{ij}(\boldsymbol{\theta})\}$ is a function of input batch $\boldsymbol{x}$, defined as:*

$$\alpha_{ij}(\boldsymbol{\theta}) := \phi'(\xi_i)\psi'(d_i^2 - d_{ij}^2) \geq 0 \tag{4}$$

*where $\phi', \psi' \geq 0$ are derivatives of $\phi, \psi$. The contrastive covariance $\mathbb{C}_\alpha[\cdot, \cdot]$ is defined as:*

$$\mathbb{C}_\alpha[\boldsymbol{a}, \boldsymbol{b}] := \sum_{i=1}^N \sum_{j\neq i} \alpha_{ij}(\boldsymbol{a}[i] - \boldsymbol{a}[j])(\boldsymbol{b}[i] - \boldsymbol{b}[j])^\top - \sum_{i=1}^N \left(\sum_{j\neq i}\alpha_{ij}\right)(\boldsymbol{a}[i] - \boldsymbol{a}[i'])(\boldsymbol{b}[i] - \boldsymbol{b}[i'])^\top \tag{5}$$

*That is,* **_minimizing_** *the loss function $\mathcal{L}_{\phi,\psi}(\boldsymbol{\theta})$ can be regarded as* **_maximizing_** *the energy function $\mathcal{E}_{\alpha=\mathrm{sg}(\alpha(\boldsymbol{\theta}))}(\boldsymbol{\theta})$ with respect to $\boldsymbol{\theta}$. Here $\mathrm{sg}(\cdot)$ means stop-gradient, i.e., the gradient of $\boldsymbol{\theta}$ is not backpropagated into $\alpha(\boldsymbol{\theta})$.*

*Proof.* By the definition of gradient descent, we have for any component $\theta$ in a high-dimensional vector $\boldsymbol{\theta}$:

$$-\frac{\partial \mathcal{L}}{\partial \theta} = -\sum_{i=1}^N \frac{\partial \boldsymbol{z}[i]}{\partial \theta}\frac{\partial \mathcal{L}}{\partial \boldsymbol{z}[i]} + \frac{\partial \boldsymbol{z}[i']}{\partial \theta}\frac{\partial \mathcal{L}}{\partial \boldsymbol{z}[i']} \tag{13}$$

Here we use the "Denominator-layout notation" and treat $\frac{\partial \mathcal{L}}{\partial \boldsymbol{z}[i]}$ as a column vector while $\frac{\partial \boldsymbol{z}[i]}{\partial \theta}$ as a row vector. Using Lemma 4, we have:

$$-\frac{\partial \mathcal{L}}{\partial \theta} = \mathbb{C}_\alpha\left[\frac{\partial \boldsymbol{z}}{\partial \theta}, \boldsymbol{z}^\top\right] \tag{14}$$

On the other hand, treating $\alpha$ as independent variables of $\boldsymbol{\theta}$, we compute (here $o_k$ is the $k$-th component of $\boldsymbol{z}$):

$$\frac{\partial \mathcal{E}_\alpha}{\partial \theta} = \frac{1}{2}\sum_k \mathbb{C}_\alpha\left[\frac{\partial o_k}{\partial \theta}, o_k\right] + \frac{1}{2}\sum_k \mathbb{C}_\alpha\left[o_k, \frac{\partial o_k}{\partial \theta}\right] \tag{15}$$

For scalar $x$ and $y$, $\mathbb{C}_\alpha[x, y] = \mathbb{C}_\alpha[y, x]$ and $\sum_k \mathbb{C}_\alpha[a_k, b_k] = \mathbb{C}_\alpha[\boldsymbol{a}, \boldsymbol{b}^\top]$ for row vector $\boldsymbol{a}$ and column vector $\boldsymbol{b}$. Therefore,

$$\frac{\partial \mathcal{E}_\alpha}{\partial \boldsymbol{\theta}} = \mathbb{C}_\alpha\left[\frac{\partial \boldsymbol{z}}{\partial \theta}, \boldsymbol{z}^\top\right] \tag{16}$$

Therefore, we have

$$\frac{\partial \mathcal{E}_\alpha}{\partial \boldsymbol{\theta}} = -\frac{\partial \mathcal{L}}{\partial \boldsymbol{\theta}} \tag{17}$$

and the proof is complete. $\qquad\square$

**Theorem 2.** *If $\psi(x) = e^{x/\tau}$, then the corresponding pairwise importance $\alpha$ (Eqn. 4) is the solution to the minimization problem:*

$$\alpha(\boldsymbol{\theta}) = \arg\min_{\alpha \in \mathcal{A}} \mathcal{E}_\alpha(\boldsymbol{\theta}) - \mathcal{R}(\alpha), \qquad \mathcal{A} := \left\{\alpha : \quad \forall i, \sum_{j\neq i}\alpha_{ij} = \tau^{-1}\xi_i\phi'(\xi_i), \ \alpha_{ij} \geq 0\right\} \tag{7}$$

*Here the regularization $\mathcal{R}(\alpha) = \mathcal{R}_H(\alpha) := \tau\sum_{i=1}^N H(\alpha_{i\cdot}) = -\tau\sum_{i=1}^N\sum_{j\neq i}\alpha_{ij}\log\alpha_{ij}$.*

*Proof.* We just need to solve the internal minimizer w.r.t. $\alpha$. Note that each $\alpha_i$ can be optimized independently.

First, we know that $\mathcal{E}_\alpha(\boldsymbol{\theta}) := \frac{1}{2}\text{tr}\mathbb{C}_\alpha[\boldsymbol{z}, \boldsymbol{z}]$ can be written as:

$$\mathcal{E}_\alpha(\boldsymbol{\theta}) = \frac{1}{2}\sum_{i \neq j} \alpha_{ij}\left[\text{tr}(\boldsymbol{z}[i] - \boldsymbol{z}[j])(\boldsymbol{z}[i] - \boldsymbol{z}[j])^\top - \text{tr}(\boldsymbol{z}[i] - \boldsymbol{z}[i'])(\boldsymbol{z}[i] - \boldsymbol{z}[i'])^\top\right] \quad (18)$$

$$= \frac{1}{2}\sum_{i \neq j} \alpha_{ij}\left[\|\boldsymbol{z}[i] - \boldsymbol{z}[j]\|_2^2 - \|\boldsymbol{z}[i] - \boldsymbol{z}[i']\|_2^2\right] \quad (19)$$

$$= \sum_{i \neq j} \alpha_{ij}\left(d_{ij}^2 - d_i^2\right) \quad (20)$$

For each $\alpha_i$, applying Lemma 5 with $c_{ij} = d_{ij}^2 - d_i^2$, the optimal solution $\alpha$ is:

$$\alpha_{ij} = \frac{1}{\tau}\exp\left(-\frac{c_{ij}}{\tau}\right)\phi'\left(\sum_{j \neq i}\exp\left(-\frac{c_{ij}}{\tau}\right)\right) \quad (21)$$

$$= \frac{1}{\tau}\exp\left(\frac{d_i^2 - d_{ij}^2}{\tau}\right)\phi'\left(\sum_{j \neq i}\exp\left(\frac{d_i^2 - d_{ij}^2}{\tau}\right)\right) \quad (22)$$

$$= \psi'(d_i^2 - d_{ij}^2)\phi'\left(\sum_{j \neq i}\psi(d_i^2 - d_{ij}^2)\right) \quad (23)$$

$$= \psi'(d_i^2 - d_{ij}^2)\phi'(\xi_i) \quad (24)$$

which coincides with Eqn. 4 that is from the gradient descent rule of the loss function $\mathcal{L}_{\phi,\psi}$.

In particular, for InfoNCE, we have $\phi(x) = \tau\log(\epsilon + x)$, $\phi'(x) = \tau/(x + \epsilon)$ and therefore:

$$\alpha_{ij} = \frac{\exp((d_i^2 - d_{ij}^2)/\tau)}{\epsilon + \sum_{j \neq i}\exp((d_i^2 - d_{ij}^2)/\tau)} = \frac{\exp(-d_{ij}^2/\tau)}{\epsilon\exp(-d_i^2/\tau) + \sum_{j \neq i}\exp(-d_{ij}^2/\tau)} \quad (25)$$

which is exactly the coefficients $\alpha_{ij}$ directly computed during minimization of $\mathcal{L}_{nce}$. If $\epsilon = 0$, then the constraint becomes $\sum_{j \neq i}\alpha_{ij} = 1$ and we have:

$$\alpha_{ij} = \frac{\exp(-d_{ij}^2/\tau)}{\sum_{j \neq i}\exp(-d_{ij}^2/\tau)} \quad (26)$$

That is, the coefficients $\alpha$ does not depend on intra-augmentation squared distance $d_i^2$. $\qquad\square$

**Corollary 1** (Contrastive Learning as Coordinate-wise Optimization). *If $\psi(x) = e^{x/\tau}$, minimizing $\mathcal{L}_{\phi,\psi}$ is equivalent to the following iterative procedure:*

$$\text{(Min-player } \alpha) \qquad \alpha_t = \arg\min_{\alpha \in \mathcal{A}}\mathcal{E}_\alpha(\boldsymbol{\theta}_t) - \mathcal{R}(\alpha) \quad (8a)$$

$$\text{(Max-player } \boldsymbol{\theta}) \qquad \boldsymbol{\theta}_{t+1} = \boldsymbol{\theta}_t + \eta\nabla_{\boldsymbol{\theta}}\mathcal{E}_{\alpha_t}(\boldsymbol{\theta}) \quad (8b)$$

*Proof.* The proof naturally follows from the conclusion of Theorem 1 and Theorem 2. $\qquad\square$

## A.2 Section 4

**Corollary 2** (Representation learning in Deep Linear CL reparameterizes Principal Component Analysis (PCA)). *When $\boldsymbol{z} = W(\boldsymbol{\theta})\boldsymbol{x}$ with a constraint $WW^\top = I$, $\mathcal{E}_\alpha$ is the objective of Principal Component Analysis (PCA) with reparameterization $W = W(\boldsymbol{\theta})$:*

$$\max_{\boldsymbol{\theta}}\mathcal{E}_\alpha(\boldsymbol{\theta}) = \frac{1}{2}\text{tr}(W(\boldsymbol{\theta})X_\alpha W^\top(\boldsymbol{\theta})) \quad \text{s.t. } WW^\top = I \quad (9)$$

*here $X_\alpha := \mathbb{C}_\alpha[\boldsymbol{x}]$ is the contrastive covariance of input $\boldsymbol{x}$.*

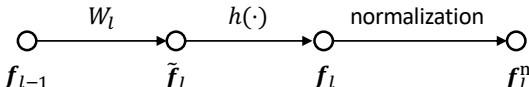

Figure 4: Notations on normalization (Sec. A.2.1).

*Proof.* Notice that in deep linear setting, $\boldsymbol{z} = W(\boldsymbol{\theta})\boldsymbol{x}$ where $W(\boldsymbol{\theta})$ does not dependent on specific samples. Therefore, $\mathbb{C}_\alpha[\boldsymbol{z}, \boldsymbol{z}] = W(\boldsymbol{\theta})\mathbb{C}_\alpha[\boldsymbol{x}, \boldsymbol{x}]W^\top(\boldsymbol{\theta}) = W(\boldsymbol{\theta})X_\alpha W^\top(\boldsymbol{\theta})$. $\square$

**Lemma 1.** *The training dynamics in* `DeepLin` *is* $\dot{W}_l = W_{>l}^\top W_{>l} W_l \mathbb{C}_\alpha[\boldsymbol{f}_{l-1}]$

*Proof.* We can start from Eqn. 13 directly and takes out $J_{>l}^\top$. This leads to

$$\dot{W}_l = J_{>l}^\top \left( \sum_{i=1}^N \frac{\partial \mathcal{L}}{\partial \boldsymbol{z}[i]} \boldsymbol{f}_{l-1}^\top[i] + \frac{\partial \mathcal{L}}{\partial \boldsymbol{z}[i']} \boldsymbol{f}_{l-1}^\top[i'] \right) = J_{>l}^\top \mathbb{C}_\alpha[\boldsymbol{z}, \boldsymbol{f}_{l-1}] \tag{27}$$

Using that $\boldsymbol{z} = J_{\geq l}\boldsymbol{f}_{l-1}$ leads to the conclusion. If the network is linear, then $J_{>l}^\top[i] = J_{>l}^\top$ is a constant. Then we can take the common factor $J_{>l}^\top J_{\geq l}$ out of the summation, yield $\dot{W}_l = J_{>l}^\top J_{\geq l} F_{l-1}$. Here $F_l := \mathbb{C}_\alpha[\boldsymbol{f}_l]$ is the contrastive covariance at layer $l$. $\square$

### A.2.1 Section 4.2

For this we talk about more general cases where the deep network is nonlinear. Let $h(\cdot)$ be the point-wise activation function and the network architecture looks like the following:

$$\boldsymbol{z}[i] := W_L h(W_{L-1}(h(\dots W_1 \boldsymbol{x}[i]))) \tag{28}$$

We consider the case where $h(\cdot)$ satisfies the following constraints:

**Definition 2** (Reversibility (Tian et al., 2020c) / Homogeneity (Du et al., 2018b)). *The activation function $h(x)$ satisfies $h(x) = h'(x)x$.*

This is satisfied by linear, ReLU, leaky ReLU and many polynomial activations (with an additional constant). With this condition, we have $\boldsymbol{f}_l[i] = D_l W_l \boldsymbol{f}_{l-1}[i]$, where $D_l = D_l(\boldsymbol{x}[i]) := \mathrm{diag}[h'(\boldsymbol{w}_{lk}^\top \boldsymbol{f}_{l-1}[i])] \in \mathbb{R}^{n_l \times n_l}$ is a diagonal matrix. For ReLU activation, the diagonal entry of $D_l$ is binary.

**Definition 3** (Reversible Layers (Tian et al., 2020c)). *A layer is reversible if there exists $J[i]$ so that $\boldsymbol{f}_{\mathrm{out}}[i] = J[i]\boldsymbol{f}_{\mathrm{in}}[i]$ and $\boldsymbol{g}_{\mathrm{in}}[i] = J^\top[i]\boldsymbol{g}_{\mathrm{out}}[i]$ for each sample $i$.*

It is clear that linear layers, ReLU and leaky ReLU are reversible. Lemma 6 tells us that $\ell_2$-normalization and LayerNorm are also reversible.

**Lemma 2.** *For MLP, if the weight $W_l$ is below a $\ell_2$-norm or LayerNorm layer, then $\frac{\mathrm{d}}{\mathrm{d}t}\|W_l\|_F^2 = 0$.*

*Proof.* See Lemma 7 that proves more general cases. $\square$

### A.2.2 Section 4.3

**Definition 4** (Aligned-rank-1 solution). *A solution $\boldsymbol{\theta} = \{W_l\}_{l=1}^L$ is called aligned-rank-1, if there exists a set of unit vectors $\{\boldsymbol{v}_l\}_{l=0}^L$ so that $W_l = \boldsymbol{v}_l \boldsymbol{v}_{l-1}^\top$ for $1 \leq l \leq L$.*

**Theorem 3** (Representation Learning with `DeepLin` is PCA). *If $\lambda_{\max}(X_\alpha) > 0$, then for any local maximum $\boldsymbol{\theta} \in \Theta$ of Eqn. 11 whose $W_{>1}^\top W_{>1}$ has distinct maximal eigenvalue:*

- *there exists a set of unit vectors $\{\boldsymbol{v}_l\}_{l=0}^L$ so that $W_l = \boldsymbol{v}_l \boldsymbol{v}_{l-1}^\top$ for $1 \leq l \leq L$, in particular, $\boldsymbol{v}_0$ is the unit eigenvector corresponding to $\lambda_{\max}(X_\alpha)$,*

- *$\boldsymbol{\theta}$ is global optimal with objective $2\mathcal{E}^* = \lambda_{\max}(X_\alpha)$.*

*Proof.* A necessary condition for $\boldsymbol{\theta}$ to be the local maximum is the critical point condition (here $\lambda_{l-1}$ is some constant):

$$W_{>l}^\top W_{>l} W_l F_{l-1} = \lambda_{l-1} W_l \tag{29}$$

Right multiplying $W_l$ on both sides of the critical point condition for $W_l$, and taking matrix trace, we have:

$$2\mathcal{E}(\boldsymbol{\theta}) = \mathrm{tr}(W_{>l}^\top W_{>l} W_l F_{l-1} W_l^\top) = \mathrm{tr}(\lambda_{l-1} W_l W_l^\top) = \lambda_{l-1} \tag{30}$$

Therefore, all $\lambda_l$ are the same, denoted as $\lambda$, and they are equal to the objective value.

Now let's consider $l = 1$. Then we have:

$$W_{>1}^\top W_{>1} W_1 X = \lambda W_1 \tag{31}$$

Applying $\mathrm{vec}(AXB) = (B^\top \otimes A)\mathrm{vec}(X)$, we have:

$$(X \otimes W_{>1}^\top W_{>1})\mathrm{vec}(W_1) = \lambda \mathrm{vec}(W_1) \tag{32}$$

with the constraint that $\|\mathrm{vec}(W_1)\|_2^2 = \|W_1\|_F^2 = 1$. Similarly, we have $2\mathcal{E}(\boldsymbol{\theta}) = \lambda$.

We then prove that $\lambda$ is the largest eigenvalue of $X \otimes W_{>1}^\top W_{>1}$. We prove by contradiction. If not, then $\mathrm{vec}(W_1)$ is not the largest eigenvector, then there is always a direction $W_1$ can move, while respecting the constraint $\|W_1\|_F = 1$ and keeping $W_{>1}$ fixed, to make $\mathcal{E}(\boldsymbol{\theta})$ strictly larger. Therefore, for any local maximum $\boldsymbol{\theta}$, $\lambda$ has to be the largest eigenvalue of $X \otimes W_{>1}^\top W_{>1}$.

Let $\{\boldsymbol{v}_{0m}\}$ be the orthonormal basis of the eigenspace of $\lambda_{\max}(X)$ and $\boldsymbol{u}$ be the (unique by the assumption) maximal unit eigenvector of $W_{>1}^\top W_{>1}$. Then $\mathrm{vec}(W_1) = \sum_m c_m \boldsymbol{v}_{0m} \otimes \boldsymbol{u}$ where $\sum_m c_m^2 = 1$, or $\mathrm{vec}(W_1) = \boldsymbol{v}_0 \otimes \boldsymbol{u}$ where the unit vector $\boldsymbol{v}_0 := \sum_m c_m \boldsymbol{v}_{0m}$. Plug $\mathrm{vec}(W_1) = \boldsymbol{v}_0 \otimes \boldsymbol{u}$ into Eqn. 32, notice that $\boldsymbol{v}_0$ is still the largest eigenvector of $X$, and we have $\lambda = \lambda_{\max}(X)\|W_{>1}\boldsymbol{u}\|_2^2$.

Now we show that $\lambda_{\max}(W_{>1}^\top W_{>1}) = \|W_{>1}\|_2^2 = 1$. If not, i.e., $\|W_{>1}\|_2 < 1$, then first by Lemma 9, we know that $\mathcal{W}_{>1} := \{W_L, W_{L-1}, \ldots, W_2\}$ must not be aligned-rank-1. Since $W_{>1}^\top W_{>1}$ is PSD and has unique maximal eigenvector $\boldsymbol{u}$, the eigenvalue associated with $\boldsymbol{u}$ must be strictly positive and thus $W_{>1}\boldsymbol{u} \neq 0$.

Then by Lemma 10, $\mathcal{W}_{>1}$ is not a local maximum of $\mathcal{J}(\mathcal{W}_{>1}; \boldsymbol{u}) := \max_{\mathcal{W}_{>1}} \|W_{>1}\boldsymbol{u}\|_2$ s.t. $\|W_l\|_F = 1$, which means that there exists $\mathcal{W}_{>1}' := \{W_L', W_{L-1}', \ldots, W_2'\}$ in the local neighborhood of $\mathcal{W}_{>1}$ so that

- $\|W_l'\|_F = 1$ for $2 \leq l \leq L$. That is, $W'$ is a feasible solution of $\mathcal{J}$.

- $\mathcal{J}(\mathcal{W}_{>1}') := \|W_{>1}'\boldsymbol{u}\|_2 > \|W_{>1}\boldsymbol{u}\|_2 = \mathcal{J}(\mathcal{W}_{>1})$.

Then let $\boldsymbol{\theta}' := \{W_L', W_{L-1}', \ldots, W_2', W_1\}$ which is a feasible solution to `DeepLin`, we have:

$$\begin{aligned}
2\mathcal{E}(\boldsymbol{\theta}') &= \mathrm{vec}^\top(W_1)(X \otimes W_{>1}'^\top W_{>1}')\mathrm{vec}(W_1) \tag{33} \\
&= (\boldsymbol{v}_0^\top \otimes \boldsymbol{u}^\top)(X \otimes W_{>1}'^\top W_{>1}')(\boldsymbol{v}_0 \otimes \boldsymbol{u}) \tag{34} \\
&= \lambda_{\max}(X)\|W_{>1}'\boldsymbol{u}\|_2^2 \tag{35} \\
&> \lambda_{\max}(X)\|W_{>1}\boldsymbol{u}\|_2^2 = \lambda = 2\mathcal{E}(\boldsymbol{\theta}) \tag{36}
\end{aligned}$$

This means that $\boldsymbol{\theta}$ is not a local maximum, which is a contradiction. Note that $\boldsymbol{\theta}'$ is not necessarily a critical point (and Eqn. 29 may not hold for $\boldsymbol{\theta}'$).

Therefore, $\lambda_{\max}(W_{>1}^\top W_{>1}) = \|W_{>1}\|_2^2 = 1$ and thus $2\mathcal{E}(\boldsymbol{\theta}) = \lambda = \lambda_{\max}(X)$.

Since $\|W_{>1}\|_2 = 1$, again by Lemma 9, $W_{L:2}$ is aligned-rank-1 and $W_{>1} = \boldsymbol{v}_L\boldsymbol{v}_1^\top$ is also a rank-1 matrix. $W_{>1}^\top W_{>1} = \boldsymbol{v}_1\boldsymbol{v}_1^\top$ has a unique maximal eigenvector $\boldsymbol{v}_1$. Therefore $\mathrm{vec}(W_1) = \boldsymbol{v}_0 \otimes \boldsymbol{v}_1$, or $W_1 = \boldsymbol{v}_1\boldsymbol{v}_0^\top$. As a result, $\boldsymbol{\theta} := \{W_{L:2}, W_1\}$ is aligned-rank-1.

Finally, since all local maxima have the same objective value $2\mathcal{E} = \lambda_{\max}(X)$, they are all global maxima. $\qquad\square$

**Remarks**. Leveraging similar proof techniques, we can also show that with BatchNorm layers, the local maxima are more constrained. From Lemma 11 we knows that if each hidden node is covered with BatchNorm, then its fan-in weights are conserved. Therefore, without loss of generality, we could set the per-filter normalization: $\|\boldsymbol{w}_{lk}\|_2 = 1$. In this case we have:

**Definition 5** (Aligned-uniform solution). *A solution $\boldsymbol{\theta}$ is called aligned-uniform, if it is aligned-rank-1, and $[\boldsymbol{v}_l]_k = \pm 1/\sqrt{n_l}$ for $1 \leq l \leq L-1$. The two end-point unit vectors ($\boldsymbol{v}_0$ and $\boldsymbol{v}_L$) can still be arbitrary.*

**Corollary 3.** *If we additionally use per-filter normalization (i.e., $\|\boldsymbol{w}_{lk}\|_2 = 1/\sqrt{n_l}$), then Thm. 3 holds and $\boldsymbol{v}_l$ is more constrained: $[\boldsymbol{v}_l]_k = \pm 1/\sqrt{n_l}$ for $1 \leq l \leq L-1$.*

*Proof.* Leveraging Lemma 12 in Theorem 3 yields the conclusion. $\qquad\square$

**Remark**. We could see that with BatchNorm, the optimization problem is more constrained, and the set of local maxima have less degree of freedom. This makes optimization better behaved.

## A.3 Section 5

**Lemma 3** (Evaluation of `ReLU2Layer`). *If Assumption 1 holds, setting $\boldsymbol{w}'_{1k} = \max(\boldsymbol{w}_{1k}, 0)$ won't change the output of `ReLU2Layer`. Furthermore, if $W_1 \geq 0$, then the formula for linear network $\mathcal{E}_\alpha = \frac{1}{2}\mathrm{tr}(W_2 W_1 X_\alpha W_1^\top W_2^\top)$ still works for `ReLU2Layer`.*

*Proof.* For the first part, we just want to prove that if Assumption 1 holds, then a 2-layer ReLU network with weights $\boldsymbol{w}_{1k}$ and $W_2$ has the same activation as another ReLU network with $\boldsymbol{w}'_{1k} = \max(\boldsymbol{w}_{1k}, 0) \geq 0$ and $W'_2 = W_2$.

We are comparing the two activations:

$$f_{1k} = \max\left(\sum_m w_{1km} x_{km}, 0\right) \tag{37}$$

$$f'_{1k} = \max\left(\sum_m \max(w_{1km}, 0) x_{km}, 0\right) = \sum_m \max(w_{1km}, 0) x_{km} \tag{38}$$

The equality is due to the fact that $\boldsymbol{x}_k \geq 0$ (by nonnegativeness). Now we consider two cases.

**Case 1**. If all $\boldsymbol{w}_{1k} \geq 0$ then obviously they are identical.

**Case 2**. If there exists $m$ so that $w_{1km} < 0$. The only situation that the difference could happen is for some specific $\boldsymbol{x}_k[i]$ so that $x_{km}[i] > 0$. By Assumption 1(one-hotness), for $m' \neq m$, $x_{km}[i] = 0$ so the gate $d_k[i] = \mathbb{I}(\boldsymbol{w}_{1k}^\top \boldsymbol{x}_k > 0) = 0$. On the other hand, $\boldsymbol{w}'^\top_{1k} \boldsymbol{x}_k = 0$ so $d'_k[i] = 0$.

Therefore, in all situations, $f_{1k} = f'_{1k}$.

For the second part, since $W_1 \geq 0$ and all input $\boldsymbol{x} \geq 0$ by non-negativeness, all gates are open and the energy $\mathcal{E}_\alpha$ of `ReLU2Layer` is the same as the linear model. $\qquad\square$

**Theorem 4** (Dynamics of `ReLU2Layer`). *If Assumption 1 holds, then the dynamics of `ReLU2Layer` with $\boldsymbol{w}_{1k} \geq 0$ is equivalent to linear dynamics with the **Sticky Weight rule**: any component that reaches 0 stays 0.*

*Proof.* Let $\boldsymbol{w}_{1k} \geq 0$ be the $k$-th filter to be considered and $w_{1km} \geq 0$ its $m$-th component. Consider a linear network with the same weights ($\boldsymbol{w}'_{1k} = \boldsymbol{w}_{1k}$ and $W'_2 = W_2$) with only the ReLU activation removed.

Now we consider the gradient rule of the ReLU network and the corresponding linear network with a sticky weight rule (here $g_k[i]$ is the backpropagated gradient sent to node $k$ for sample $i$, and $d_k[i]$ is the binary gating for sample $i$ at node $k$):

$$\dot{w}_{1km} = \sum_i g_k[i] d_k[i] x_m[i] \tag{39}$$

$$\dot{w}'_{1km} = \mathbb{I}(w_{1km} > 0) \sum_i g'_k[i] x_m[i] \tag{40}$$

Thanks to Lemma 13, we know the forward pass between two networks are identical and thus $g_k[i] = g'_k[i]$ so we don't need to consider the difference between backpropagated gradient.

In the following, we will show that each summand of the two equations is identical.

**Case 1.** $x_m[i] = 0$. In that case, $g_k[i]x_m[i] = g_k[i]d_k[i]x_m[i] = 0$ regardless of whether the gate $d_k[i]$ is open or closed.

**Case 2.** $x_m[i] > 0$. There are two subcases:

*Subcase 1:* $d_k[i] = 1$. In this case, the ReLU gating of $k$-th filter is open, then $g'_k[i]x_m[i] = g_k[i]x_m[i] = g_k[i]d_k[i]x_m[i]$. By Assumption 1(One-hotness), for other $m' \neq m$, $x_{km'}[i] = 0$, since $d_k[i] = 1$, it must be the case that $w_{1km} > 0$ and thus $\mathbb{I}(w_{1km} > 0) = 1$. So the two summands are identical.

*Subcase 2:* $d_k[i] = 0$. Then $w_{1km}$ must be 0, otherwise since $\boldsymbol{x}_k \geq 0$ (nonnegativeness), we have $\boldsymbol{w}_{1k}^\top \boldsymbol{x}_k[i] \geq w_{1km}x_m[i] > 0$ and the gating of $k$-th filter must open. Therefore, the two summands are both 0: the ReLU one is because $d_k[i] = 0$ and the linear one is due to $\mathbb{I}(w_{1km} > 0) = 0$. □

**Theorem 5.** *If Assumption 1 holds, then in* `ReLU2Layer1Hid`*, $\boldsymbol{w}_1 \to \boldsymbol{e}_m$ for certain $m$.*

*Proof.* In `ReLU2Layer1Hid`, since there is only one node, we have $X = \mathbb{C}_\alpha[\boldsymbol{x}_1, \boldsymbol{x}_1] = \mathbb{C}_\alpha[\boldsymbol{x}, \boldsymbol{x}]$. By Theorem 4, the dynamics of $\boldsymbol{w}_1$ is the linear dynamics plus the sticky weight rule, which is:

$$\dot{\boldsymbol{w}}_1 = \mathrm{diag}(\boldsymbol{w}_1 > 0)X\boldsymbol{w}_1 \tag{41}$$

By Lemma 3, the negative parts of $\boldsymbol{w}_1$ can be removed without changing the result. Let's only consider the nonnegative part of $\boldsymbol{w}$ and remove corresponding rows and columns of $X$.

Note that the linear dynamics $\dot{\boldsymbol{w}}_1 = X\boldsymbol{w}_1$ will converge to certain maximal eigenvector $\boldsymbol{y}$ (or its scaled version, depending on whether we have norm constraint or not). By Lemma 14, as long as $X$ is not a scalar, $\boldsymbol{y}$ has at least one negative entry. Therefore, by continuity of the trajectory of the linear dynamics, from $\boldsymbol{w}_1$ to $\boldsymbol{y}$, the trajectory must cross the boundary of the polytope $\boldsymbol{w}_1 \geq 0$ that require all entries to be nonnegative.

After that, according to the sticky weight rule, in the ReLU dynamics, the corresponding component (say $w_{1m}$) stays at zero. We can remove the corresponding $m$-th row and column of $X$, and the process repeats until $X$ becomes a scalar. Then $\boldsymbol{w}_1$ converges to that remaining dimension. Since $\boldsymbol{w}_1 \geq 0$, it must be the case that $\boldsymbol{w}_1 \to \boldsymbol{e}_m$ for some $m$. □

**Theorem 6** (`ReLU2Layer` encourages diversity). *If Assumption 1 holds, then for any local optimal $(W_2, W_1) \in \Theta$ of `ReLU2Layer` with $\mathcal{E} > 0$, either $W_1 = \boldsymbol{v}\boldsymbol{e}_m^\top$ for some $m$ and $\boldsymbol{v} \geq 0$, or $\mathrm{rank}(W_1) > 1$.*

*Proof.* We just need to prove that if the local optimal solution $(W_2, W_1)$ satisfies $\mathrm{rank}(W_1) = 1$, then $W_1 = \boldsymbol{v}\boldsymbol{e}_m^\top$ for some $m$ and $\boldsymbol{v} \geq 0$.

Since $\mathrm{rank}(W_1) = 1$ and $\|W_1\|_F = 1$, by Lemma 8 we know that there exists unit vectors $\boldsymbol{u}$ and $\boldsymbol{v}$ so that $W_1 = \boldsymbol{v}\boldsymbol{u}^\top$. Since $W_1 \geq 0$, we can pick $\boldsymbol{u} \geq 0$ and $\boldsymbol{v} \geq 0$. Otherwise if $\boldsymbol{u}$ has both positive and negative elements, then picking any nonzero element of $\boldsymbol{v}$, the corresponding rows/colums of $W_1$ will also have both signs, which is a contradiction.

Note that the objective function is

$$2\mathcal{E} = \mathrm{tr}(W_2 F_1 W_2^\top) = \mathrm{tr}(W_2 W_1 X_\alpha W_1^\top W_2^\top) = (\boldsymbol{u}^\top X_\alpha \boldsymbol{u})\|W_2 \boldsymbol{u}\|_2^2 > 0 \tag{42}$$

Therefore, $\boldsymbol{u}^\top X_\alpha \boldsymbol{u} > 0$ and $\|W_2 \boldsymbol{u}\|_2 > 0$. By Lemma 10, we know that if $W_2$ with the constraint $\|W_2\|_F = 1$ is an local optimal, $W_2$ is a rank-1 matrix with decomposition $W_2 = \boldsymbol{b}\boldsymbol{v}^\top$ with $\|\boldsymbol{b}\|_2 = 1$.

Then we have $2\mathcal{E} = \boldsymbol{u}^\top X_\alpha \boldsymbol{u} > 0$ with $\boldsymbol{u} \geq 0$. From the proof of Lemma 14, we know that $X_\alpha$ has a unique *minimal* all-positive eigenvector $\boldsymbol{c} > 0$.

If there are $\geq 2$ positive elements in $\boldsymbol{u}$, then we can always create a vector $\boldsymbol{a}$ (with mixed signs in its elements) so that (1) $\boldsymbol{a}$ has the same non-zero support as $\boldsymbol{u}$ and (2) $\boldsymbol{a}^\top \boldsymbol{c} = 0$. Therefore, $\boldsymbol{a}$ is in the space of orthogonal complement of $\boldsymbol{c}$. Since $\boldsymbol{c}$ is the unique minimal eigenvector, moving $\boldsymbol{u}$ along the direction of $\boldsymbol{a}$ will strictly improve $\mathcal{E}$, which contradicts with the fact that $(W_2, W_1)$ is locally optimal.

| Dataset | Methods | 100 epochs | 300 epochs | 500 epochs |
|---------|---------|------------|------------|------------|
| CIFAR-10 | $\mathcal{L}_{nce}$ | $86.84 \pm 0.26$ | $89.19 \pm 0.15$ | $\mathbf{91.07 \pm 0.12}$ |
| | $\alpha$-CL-direct (Eqn. 43) | $87.74 \pm 0.28$ | $89.76 \pm 0.26$ | $91.06 \pm 0.09$ |
| | $\alpha$-CL-direct (Eqn. 44) | $\mathbf{87.91 \pm 0.12}$ | $\mathbf{89.89 \pm 0.18}$ | $91.06 \pm 0.17$ |
| CIFAR-100 | $\mathcal{L}_{nce}$ | $60.70 \pm 0.40$ | $64.22 \pm 0.19$ | $\mathbf{66.84 \pm 0.16}$ |
| | $\alpha$-CL-direct (Eqn. 43) | $63.28 \pm 0.31$ | $65.71 \pm 0.20$ | $66.73 \pm 0.13$ |
| | $\alpha$-CL-direct (Eqn. 44) | $\mathbf{63.47 \pm 0.06}$ | $\mathbf{65.86 \pm 0.24}$ | $66.57 \pm 0.21$ |
| STL10 | $\mathcal{L}_{nce}$ | $82.09 \pm 0.31$ | $86.96 \pm 0.19$ | $87.31 \pm 0.17$ |
| | $\alpha$-CL-direct (Eqn. 43) | $83.00 \pm 0.28$ | $87.35 \pm 0.28$ | $87.63 \pm 0.29$ |
| | $\alpha$-CL-direct (Eqn. 44) | $\mathbf{83.20 \pm 0.17}$ | $\mathbf{87.36 \pm 0.12}$ | $\mathbf{87.71 \pm 0.14}$ |

Table 3: Top-1 downstream task accuracy with ResNet50 backbone and 256 batchsize. Learning rate is 0.001. We also compare unnormalized $\alpha$-CL-direct (Eqn. 43) versus (normalized) $\alpha$-CL-direct (Eqn. 44). Normalized version, which is used in the main text of the paper, performs slightly better.

| Exponent $p$ | $p = 2$ | $p = 4$ | $p = 6$ | $p = 8$ | $p = 10$ |
|--------------|---------|---------|---------|---------|----------|
| Top-1 accuracy (500 epochs) | $83.74 \pm 0.18$ | $84.06 \pm 0.24$ | $\mathbf{84.08 \pm 0.42}$ | $83.91 \pm 0.28$ | $83.56 \pm 0.13$ |

Table 4: Ablation study on different exponent $p$ in STL10 for the normalized pairwise importance (Eqn. 44) in $\alpha$-CL-direct.

Therefore, the unit vector $\boldsymbol{u}$ has only 1 positive entry, which is $\boldsymbol{e}_m$ for some $m$. Fig. 3 shows one example of learned weights with rank $> 1$. □

# B More Experiments

We also provide experiments with different batchsize (i.e., 256) and ablation studies on different exponent $p$ in the direct version of $\alpha$-CL. Note that we refer an unnormalized $\alpha$-CL-direct as the following:

$$\alpha_{ij} = \exp(-d_{ij}^p/\tau) \tag{43}$$

while (normalized) $\alpha$-CL-direct as the following (same as Eqn. 12 in the main text):

$$\alpha_{ij} = \frac{\exp(-d_{ij}^p/\tau)}{\sum_j \exp(-d_{ij}^p/\tau)} \tag{44}$$

By default, we set the exponent $p = 4$ and $\tau = 0.5$.

# C Other Lemmas

**Lemma 4** (Gradient Formula of contrastive Loss (Eqn. 1) (extension of Lemma 2 in (Jing et al., 2022)). *Consider the loss function*

$$\min_{\boldsymbol{\theta}} \mathcal{L}_{\phi,\psi}(\boldsymbol{\theta}) := \sum_{i=1}^{N} \phi\left(\sum_{j \neq i} \psi(d_i^2 - d_{ij}^2)\right) \tag{45}$$

*Then for any matrix (or vector) variable A, we have:*

$$\sum_{i=1}^{N} \frac{\partial \mathcal{L}_{\phi,\psi}}{\partial \boldsymbol{z}[i]} A^\top[i] + \frac{\partial \mathcal{L}_{\phi,\psi}}{\partial \boldsymbol{z}[i']} A^\top[i'] = -\mathbb{C}_\alpha[\boldsymbol{z}, A] \tag{46}$$

*and*

$$\sum_{i=1}^{N} A[i] \frac{\partial \mathcal{L}_{\phi,\psi}}{\partial \boldsymbol{z}[i]} + A[i'] \frac{\partial \mathcal{L}_{\phi,\psi}}{\partial \boldsymbol{z}[i']} = -\mathbb{C}_\alpha[A, \boldsymbol{z}^\top] \tag{47}$$

*where $\mathbb{C}_\alpha[\cdot, \cdot]$ is the* contrastive covariance *defined as (here $\beta_i := \sum_{j \neq i} \alpha_{ij}$):*

$$\mathbb{C}_\alpha[\boldsymbol{x}, \boldsymbol{y}] := \sum_{i,j=1}^{N} \alpha_{ij}(\boldsymbol{x}[i] - \boldsymbol{x}[j])(\boldsymbol{y}[i] - \boldsymbol{y}[j])^\top - \sum_{i=1}^{N} \beta_i(\boldsymbol{x}[i] - \boldsymbol{x}[i'])(\boldsymbol{y}[i] - \boldsymbol{y}[i'])^\top \tag{48}$$

*and $\alpha$ is defined as the following:*

$$\alpha_{ij} := \phi'\left(\sum_{j \neq i} \psi(d_i^2 - d_{ij}^2)\right)\psi'(d_i^2 - d_{ij}^2) \geq 0 \qquad (49)$$

*where $\phi', \psi'$ are derivatives of $\phi, \psi$.*

*Proof.* Taking derivative of the loss function $\mathcal{L} = \mathcal{L}_{\phi,\psi}$ w.r.t. $z[i]$ and $z[i']$, we have:

$$\frac{\partial \mathcal{L}}{\partial z[i]} = \sum_{j \neq i} \alpha_{ij}(z[j] - z[i']) + \sum_{j \neq i} \alpha_{ji}(z[j] - z[i]) \qquad (50)$$

$$\frac{\partial \mathcal{L}}{\partial z[i']} = \sum_{j \neq i} \alpha_{ij}(z[i'] - z[i]) = \beta_i(z[i'] - z[i]) \qquad (51)$$

We just need to check the following:

$$\sum_i \left(\sum_{j \neq i} \alpha_{ij}(z[j] - z[i']) + \sum_{j \neq i} \alpha_{ji}(z[j] - z[i])\right) A^\top[i] + \sum_i \beta_i(z[i'] - z[i]) A^\top[i'] \qquad (52)$$

To see this, we only need to check whether the following is true:

$$-\Sigma_0 = \sum_i \left(\sum_{j \neq i} \alpha_{ij}(z[j] - z[i']) + \sum_{j \neq i} \alpha_{ji}(z[j] - z[i])\right) A^\top[i] + \sum_i \beta_i(z[i'] - z[i]) A^\top[i] \qquad (53)$$

which means that

$$-\Sigma_0 = \sum_i \left(\sum_{j \neq i} \alpha_{ij}(z[j] - z[i]) + \sum_{j \neq i} \alpha_{ji}(z[j] - z[i])\right) A^\top[i] \qquad (54)$$

Since $\alpha_{ii}(z[i] - z[i]) = 0$ for arbitrarily defined $\alpha_{ii}$, $j$ can also take the value of $i$, this leads to

$$-\Sigma_0 = \sum_{i,j} \alpha_{ij}(z[j] - z[i]) A^\top[i] + \sum_{i,j} \alpha_{ji}(z[j] - z[i]) A^\top[i] \qquad (55)$$

Swapping indices for the second term, we have:

$$-\Sigma_0 = \sum_{i,j} \alpha_{ij}(z[j] - z[i]) A^\top[i] + \sum_{i,j} \alpha_{ij}(z[i] - z[j]) A^\top[j] \qquad (56)$$

$$= \sum_{i,j} \alpha_{ij}(z[j] - z[i]) A^\top[i] - \sum_{i,j} \alpha_{ij}(z[j] - z[i]) A^\top[j] \qquad (57)$$

$$= -\sum_{i,j} \alpha_{ij}(z[j] - z[i])(A^\top[j] - A^\top[i]) \qquad (58)$$

and the conclusion follows. $\qquad \square$

**Lemma 5.** *The following minimization problem:*

$$\min_{p_j} \sum_j c_j p_j - \tau H(p) \quad \text{s.t.} \sum_j p_j = \frac{1}{\tau} x_0 \phi'(x_0) \qquad (59)$$

*where $H(p) := -\sum_j p_j \log p_j$ is the entropy and $x_0 := \sum_j e^{-c_j/\tau}$, has close-form solution:*

$$p_j = \frac{1}{\tau} \exp(-c_j/\tau)\phi'\left(\sum_j \exp(-c_j/\tau)\right) \qquad (60)$$

*Proof.* Define the following Lagrangian multiplier:

$$J(\alpha, \boldsymbol{\theta}) := \sum_j c_j p_j - \tau H(p) + \mu \left( \sum_j p_j - \frac{1}{\tau} x_0 \phi'(x_0) \right) \tag{61}$$

Taking derivative w.r.t $p_j$ and we have:

$$\frac{\partial J}{\partial p_j} = c_j + \tau (\log p_j + 1) - \mu = 0 \tag{62}$$

which gives the solution

$$p_j = \exp\left(\frac{\mu}{\tau} - 1\right) \exp\left(-\frac{c_j}{\tau}\right) := Z \exp\left(-\frac{c_j}{\tau}\right) \tag{63}$$

where $Z$ can be computed via the constraint:

$$Z = \frac{1}{\tau} \frac{x_0 \phi'(x_0)}{\sum_j e^{-c_j/\tau}} = \frac{1}{\tau} \phi'(x_0) \tag{64}$$

$\square$

**Lemma 6.** *The normalization function* $\boldsymbol{y} = (\boldsymbol{x} - \mathrm{mean}(\boldsymbol{x}))/\|\boldsymbol{x}\|_2$ *has the following forward/backward rule:*

$$\boldsymbol{y} = J(\boldsymbol{x})\boldsymbol{x}, \qquad \frac{\partial \boldsymbol{y}}{\partial \boldsymbol{x}} = J^\top(\boldsymbol{x}) \tag{65}$$

*where* $J(\boldsymbol{x}) := \frac{1}{\|P_1^\perp \boldsymbol{x}\|_2} P_{\boldsymbol{x},1}^\perp$ *is a symmetric matrix. For* $\boldsymbol{y} = \boldsymbol{x}/\|\boldsymbol{x}\|_2$, *the relationship still holds with* $J(\boldsymbol{x}) = \frac{1}{\|\boldsymbol{x}\|_2} P_{\boldsymbol{x}}^\perp$.

*Proof.* See Theorem 5 in (Tian, 2018). $\square$

**Lemma 7.** *Suppose the output of a linear layer (with a weight matrix $W_l$) connects to a $\ell_2$ regularization or LayerNorm through reversible layers, then $\frac{\mathrm{d}}{\mathrm{dt}} \|W_l\|_F^2 = 0$.*

*Proof.* From Lemma, for each sample $i$, we have its gradient before/after the normalization layer (say it is layer $m$) to be the following:

$$\boldsymbol{g}_m[i] = J_m^{\mathrm{n}}[i]^\top \boldsymbol{g}_m^{\mathrm{n}}[i] \tag{66}$$

where $\boldsymbol{g}_m[i]$ is the gradient after back-propagating through normalization, and $\boldsymbol{g}_m^{\mathrm{n}}[i]$ is the gradient sending from the top level.

Here $J_m^{\mathrm{n}}[i] = \frac{1}{\|P_1^\perp \boldsymbol{f}_m[i]\|_2} P_{\boldsymbol{f}_m[i],1}^\perp$ for LayerNorm and $J_m^{\mathrm{n}}[i] = \frac{1}{\|\boldsymbol{f}_m[i]\|_2} P_{\boldsymbol{f}_m[i]}^\perp$ for $\ell_2$ normalization. For $W_l$, its gradient update rule is:

$$\dot{W}_l = \sum_i \tilde{\boldsymbol{g}}_l[i] \boldsymbol{f}_{l-1}^\top[i] \tag{67}$$

By reversibility, we know that $\tilde{\boldsymbol{g}}_l[i] = J_{(\tilde{l},m]}^\top[i] \boldsymbol{g}[i]$, where $J_{(\tilde{l},m]}[i]$ is the Jacobian after the linear layer $\tilde{l}$ till layer $m$, right before the normalization layer. Therefore, we have:

$$\mathrm{tr}(W_l^\top \dot{W}_l) = \sum_i \mathrm{tr}(W_l^\top J_{(\tilde{l},m]}^\top[i] J_m^{\mathrm{n}}[i]^\top \boldsymbol{g}_m^{\mathrm{n}}[i] \boldsymbol{f}_{l-1}^\top[i]) \tag{68}$$

$$= \sum_i \mathrm{tr}(\boldsymbol{f}_{l-1}^\top[i] W_l^\top J_{(\tilde{l},m]}^\top[i] J_m^{\mathrm{n}}[i]^\top \boldsymbol{g}_m^{\mathrm{n}}[i]) \tag{69}$$

$$= \sum_i \mathrm{tr}(\boldsymbol{f}_m^\top[i] J_m^{\mathrm{n}}[i]^\top \boldsymbol{g}_m^{\mathrm{n}}[i]) \tag{70}$$

$$= 0 \tag{71}$$

The last two equality is due to reversibility $\boldsymbol{f}_m[i] = J_{(\tilde{l},m]}[i] W_l \boldsymbol{f}_{l-1}[i]$ and the property of normalization layers: $J_m^{\mathrm{n}}[i] \boldsymbol{f}_m[i] = 0$, since a vector projected to its own complementary space is always zero $P_{\boldsymbol{f}_m[i]}^\perp \boldsymbol{f}_m[i] = 0$.

Then we have

$$\frac{d}{dt}\|W_l\|_F^2 = \frac{d}{dt}\text{tr}(W_l^\top W_l) = \text{tr}(\dot{W}_l^\top W_l) + \text{tr}(W_l^\top \dot{W}_l) = 0 \tag{72}$$

□

**Lemma 8.** *For every rank-1 matrix $A$ with $\|A\|_F = 1$, there exists $\|\boldsymbol{u}\|_2 = \|\boldsymbol{v}\|_2 = 1$ so that $A = \boldsymbol{u}\boldsymbol{v}^\top$.*

*Proof.* Since $A$ is rank-1, it is clear that there exists $\boldsymbol{u}'$ and $\boldsymbol{v}'$ so that $A = \boldsymbol{u}'\boldsymbol{v}'^\top$. Since $\|A\|_F = 1$, we have $\|A\|_F^2 := \text{tr}(AA^\top) = \|\boldsymbol{u}'\|_2^2\|\boldsymbol{v}'\|_2^2 = 1$. Therefore, taking $\boldsymbol{u} = \boldsymbol{u}'/\|\boldsymbol{u}'\|_2$ and $\boldsymbol{v} = \boldsymbol{v}'/\|\boldsymbol{v}'\|_2$, we have $A = \boldsymbol{u}\boldsymbol{v}^\top$. □

**Lemma 9.** *If $\|W_l\|_F = 1$ for $1 \leq l \leq L$, then $\|W_L W_{L-1} \ldots W_1\|_2 = 1$ if any only if $W_L, W_{L-1}, \ldots, W_1$ are aligned-rank-1 (Def. 4).*

*Proof.* If $W_L, W_{L-1}, \ldots, W_1$ are aligned-rank-1, then by its definition, there exists unit vectors $\{\boldsymbol{v}_l\}_{l=0}^L$ so that $W_l = \boldsymbol{v}_l\boldsymbol{v}_{l-1}^\top$. Therefore, $\|W_L W_{L-1} \ldots W_1\|_2^2 = \|\boldsymbol{v}_L\boldsymbol{v}_0^\top\|_2^2 = \lambda_{\max}(\boldsymbol{v}_L\boldsymbol{v}_0^\top \boldsymbol{v}_0\boldsymbol{v}_L^\top) = \lambda_{\max}(\boldsymbol{v}_L\boldsymbol{v}_L^\top) = 1$.

Then we prove the other direction. Note that

$$\|W_L W_{L-1} \ldots W_1\|_2 \leq \prod_{l=1}^L \|W_l\|_2 \leq \prod_{l=1}^L \|W_l\|_F = 1 \tag{73}$$

and the equality only holds when all $W_l$ are rank-1. By Lemma 8, for any $l$, there exists unit vectors $\boldsymbol{v}_l'$, $\boldsymbol{v}_{l-1}$ so that $W_l = \boldsymbol{v}_l'\boldsymbol{v}_{l-1}^\top$. To show that they must be aligned (i.e. $\boldsymbol{v}_l = \pm\boldsymbol{v}_l'$), we prove by contradiction.

Suppose $\|W_L W_{L-1} \ldots W_1\|_2 = 1$ but for some $l$, $\boldsymbol{v}_l' \neq \pm\boldsymbol{v}_l$ and thus $|\boldsymbol{v}_l^\top \boldsymbol{v}_l'| < 1$. Then $W_{l+1}W_l = (\boldsymbol{v}_l^\top \boldsymbol{v}_l')\boldsymbol{v}_{l+1}\boldsymbol{v}_{l-1}^\top$ and $\|W_{l+1}W_l\|_2 \leq \|W_{l+1}W_l\|_F = |\boldsymbol{v}_l^\top \boldsymbol{v}_l'| < 1$. Therefore, $\|W_L W_{L-1} \ldots W_1\|_2 < 1$, which is a contradiction.

Note that for $W_l = \pm\boldsymbol{v}_l\boldsymbol{v}_{l-1}^\top$, we can always move around the signs to either $\boldsymbol{v}_0$ or $\boldsymbol{v}_L$ to fit into the definition of aligned-rank-1. □

**Lemma 10.** *For the following optimization problem with a given fixed vector $\boldsymbol{u} \neq 0$:*

$$\max_{\mathcal{W}} \mathcal{J}(\mathcal{W}; \boldsymbol{u}) := \|W_L W_{L-1} \ldots W_1 \boldsymbol{u}\|_2 \quad \text{s.t. } \|W_l\|_F = 1, \tag{74}$$

*where $\mathcal{W} = \{W_L, W_{L-1}, \ldots, W_1\}$. If $\mathcal{W}^*$ is a local maximum solution (i.e., there exists a neighborhood $\mathcal{N}(\mathcal{W}^*)$ of $\mathcal{W}^*$ so that for any $\mathcal{W} \in \mathcal{N}(\mathcal{W}^*)$, $\mathcal{J}(\mathcal{W}) \leq \mathcal{J}(\mathcal{W}^*)$), and $\mathcal{J}(\mathcal{W}^*) > 0$, then $\mathcal{W}^*$ is an aligned-rank-1 solution (Def. 4).*

*Proof.* Let $\boldsymbol{v}_{L-1}' := W_{L-1}^* W_{L-2}^* \ldots W_1^* \boldsymbol{u}$. Note that $\boldsymbol{v}_{L-1}' \neq 0$ (otherwise $\mathcal{J}(\mathcal{W}^*)$ would be zero). Consider the following optimization subproblem (here we optimize over $W_L$ and treat $\boldsymbol{v}_{L-1}'$ as a fixed vector).

$$\max_{W_L} \mathcal{J}(W_L; W_{-L}^*) = \|W_L \boldsymbol{v}_{L-1}'\|_2 \quad \text{s.t. } \|W_L\|_F = 1 \tag{75}$$

By local optimality of $\mathcal{W}^*$, $W_L^*$ must be the local maximum of Eqn. 75 and thus a critical point, since both the objective and the constraints are differentiable. Note that $\|W_L \boldsymbol{v}_{L-1}'\|_2$ is a vector 2-norm and all critical points of Eqn. 75 must satisfy

$$W_L \boldsymbol{v}_{L-1}' \boldsymbol{v}_{L-1}'^\top = \lambda W_L \tag{76}$$

for some constant $\lambda$. Notice that to satisfy this condition, each row of $W_L$ must be an eigenvector of $\boldsymbol{v}_{L-1}' \boldsymbol{v}_{L-1}'^\top$. For a solution to be local maximal, $\lambda$ is the largest eigenvalue of $\boldsymbol{v}_{L-1}' \boldsymbol{v}_{L-1}'^\top$, and each row of $W_L$ is the corresponding eigenvector. It is clear that the rank-1 matrix $\boldsymbol{v}_{L-1}' \boldsymbol{v}_{L-1}'^\top$ has a unique maximum eigenvalue $\|\boldsymbol{v}_{L-1}'\|_2^2 > 0$ with its corresponding one-dimensional eigenspace span by $\boldsymbol{v}_{L-1} := \boldsymbol{v}_{L-1}'/\|\boldsymbol{v}_{L-1}'\|_2$ (while all other eigenvalues are zeros). Therefore, $W_L^*$ as the local maximum of Eqn. 75, must have:

$$W_L^* = \boldsymbol{v}_L \boldsymbol{v}_{L-1}^\top \tag{77}$$

for some $\|\boldsymbol{v}_L\|_2 = 1$.

Now let $\boldsymbol{v}'_{L-2} := W^*_{L-2}\ldots W^*_1 \boldsymbol{u}$. Similarly, $\boldsymbol{v}'_{L-2} \neq 0$ (otherwise $\mathcal{J}(\mathcal{W}^*)$ would be zero). Then $\boldsymbol{v}'_{L-1} = W^*_{L-1} \boldsymbol{v}'_{L-2}$. Treating $\boldsymbol{v}'_{L-2}$ as a fixed vector and varying $W_{L-1}$ and $W_L$ simultaneously, then since $W^*_{L:1}$ is a local maximal solution, $W^*_L$ must take the form of Eqn. 77 given any $W_{L-1}$, which means that the objective function now becomes

$$\mathcal{J}(W_{L-1}; W^*_{-(L-1)}) = \|W^*_L \boldsymbol{v}'_{L-1}\|_2 = \|\boldsymbol{v}_L \boldsymbol{v}^\top_{L-1} \boldsymbol{v}'_{L-1}\|_2 = \|\boldsymbol{v}'_{L-1}\|_2 = \|W_{L-1}\boldsymbol{v}'_{L-2}\|_2 \quad (78)$$

and the subproblem becomes:

$$\max_{W_{L-1}} \|W_{L-1}\boldsymbol{v}'_{L-2}\|_2 \quad \text{s.t.} \ \|W_{L-1}\|_F = 1 \tag{79}$$

Repeating this process, we know $W^*_{L-1}$ must satisfy:

$$W^*_{L-1} = \boldsymbol{v}_{L-1}\boldsymbol{v}^\top_{L-2} \tag{80}$$

for $\boldsymbol{v}_{L-2} := \boldsymbol{v}'_{L-2}/\|\boldsymbol{v}'_{L-2}\|_2$. This procedure can be repeated until $W_1$ and the prove is complete. $\qquad \square$

**Lemma 11.** $\frac{\mathrm{d}}{\mathrm{d}t}\|\boldsymbol{w}_k\|_2^2 = 0$, if node $k$ is under BatchNorm.

*Proof.* For BN, it is a layer with reversibility on each filter $k$. We use $\boldsymbol{f}_k, \boldsymbol{g}_k \in \mathbb{R}^N$ to represent the activation/gradient at node $k$ in a batch of size $N$. The forward/backward operation of BN can be written as:

$$\boldsymbol{f}^\mathrm{n}_k = J_k \boldsymbol{f}_k, \quad \boldsymbol{g}_k = J^\top_k \boldsymbol{g}^\mathrm{n}_k \tag{81}$$

Here $J_k = J^\top_k = \frac{1}{\|P^\perp_\mathbf{1} \boldsymbol{f}_k\|_2} P^\perp_{\boldsymbol{f}_k,\mathbf{1}}$ is the Jacobian matrix at each node $k$.

We check how the weight $\boldsymbol{w}_k$ changes under BatchNorm. Here we have $\boldsymbol{f}_k = h(F_{l-1}\boldsymbol{w}_k)$ where $h$ is a reversible activation and $F_{l-1} \in \mathbb{R}^{N \times n_{l-1}}$ contains all output from the last layer. Then we have:

$$\dot{\boldsymbol{w}}_k = \sum_i h'_i g_k[i] \boldsymbol{f}_{l-1}[i] = F^\top_{l-1} D_k \boldsymbol{g}_k = F^\top_{l-1} D_k J^\top_k \boldsymbol{g}^\mathrm{n}_k \tag{82}$$

where $D_k := \mathrm{diag}([h'_i]^N_{i=1}) \in \mathbb{R}^{N \times N}$. Due to reversibility, we have $\boldsymbol{f}_k = h(F_{l-1}\boldsymbol{w}_k) = D_k F_{l-1}\boldsymbol{w}_k$. Therefore,

$$\boldsymbol{w}^\top_k \dot{\boldsymbol{w}}_k = \boldsymbol{w}^\top_k F^\top_{l-1} D_k J^\top_k \boldsymbol{g}^\mathrm{n}_k = \boldsymbol{f}^\top_k J^\top_k \boldsymbol{g}^\mathrm{n}_k = 0 \tag{83}$$

$\qquad \square$

**Lemma 12** (BatchNorm regularization). *Consider the following optimization problem with a fixed vector $\boldsymbol{u} \neq 0$:*

$$\max_{\mathcal{W}} \mathcal{J}(\mathcal{W}) := \|W_L W_{L-1}\ldots W_1 \boldsymbol{u}\|_2 \quad \text{s.t.} \ \|W_L\|_F = 1, \quad \|\boldsymbol{w}_{lk}\|_2 = 1/\sqrt{n_l} \tag{84}$$

*where $\mathcal{W} := \{W_L, W_{L-1}, \ldots, W_1\}$ and $\boldsymbol{w}_{lk}$ are rows of $W_l$ (i.e., weight of the $k$-th filter at layer $l$). Then Lemma 10 still holds by replacing aligned-ranked-one with aligned-uniform condition.*

*Proof.* The proof is basically the same. The only difference here is that the sub-problem (Eqn. 79) becomes:

$$\max_{W_l} \|W_l \boldsymbol{v}'_{l-1}\|_2 \quad \text{s.t.} \ \|\boldsymbol{w}_{lk}\|_2 = 1/\sqrt{n_l} \tag{85}$$

for $1 \leq l \leq L-1$. The critical point condition now becomes (here $\Lambda$ is a diagonal matrix):

$$W_l \boldsymbol{v}'_{l-1} \boldsymbol{v}'^\top_{l-1} = \Lambda W_l \tag{86}$$

That is, each row of $W_l$ now has a different constant. Since the eigenvalue of $\boldsymbol{v}'_{l-1}\boldsymbol{v}'^\top_{l-1}$ can only be 0 or 1, and 0 won't work (otherwise the corresponding row of $W_l$ would be a zero vector, violating the row-norm constraint), all diagonal element of $\lambda$ has to be 1. Therefore, $W_l = \boldsymbol{v}_l \boldsymbol{v}^\top_{l-1}$. Due to row-normalization, we have $[\boldsymbol{v}_l]_k = \pm 1/\sqrt{n_l}$ for $1 \leq l \leq L-1$, while $\boldsymbol{v}_L$ and $\boldsymbol{v}_0$ can still take arbitrary unit vector. $\qquad \square$

**Lemma 13.** *If Assumption 1(Nonnegativeness) holds, then a 2-layer ReLU network with weights $\boldsymbol{w}_{1k} \geq 0$ and $W_2$ has the same activations (i.e., $\boldsymbol{f}_l = \boldsymbol{f}'_l$) as its linear network counterpart with the same weights $\boldsymbol{w}'_{1k} = \boldsymbol{w}_{1k}$ and $W'_2 = W_2$.*

*Proof.* Since $W'_2 = W_2$, we only need to prove $\boldsymbol{f}_1 = \boldsymbol{f}'_1$. For each filter $k$, we have its activation $f_{1k} = \max(\sum_m w_{1km}x_{km}, 0)$ and $f'_{1k} = \sum_m w'_{1km}x_{km} = \sum_m w_{1km}x_{km}$. By Assumption 1(non-negativeness), all $x_{km} \geq 0$. Since $w_{1km} \geq 0$, $\sum_m w_{1km}x_{km} \geq 0$ and $f_{1k} = f'_{1k}$. $\square$

**Lemma 14.** *If Assumption 1 holds, $M \geq 2$, $\boldsymbol{x}_1$ covers all $M$ modes, and $\alpha_{ij} > 0$, then the maximal eigenvector of $X_\alpha$ always contains at least one negative entry.*

*Proof.* Let $X_k := \mathbb{C}_\alpha[\boldsymbol{x}_k, \boldsymbol{x}_k]$. By Lemma 15, all off-diagonal elements of $X_k$ are negative. Then $X_k$ can be written as $X_k = \beta I - X'_k$ for some $\beta$ where $X'_k$ is a symmetric matrix whose entries are all positive. By Perron–Frobenius theorem, $X'_k$ has a unique maximal eigenvector $\boldsymbol{u}_k > 0$ (with all positive entries) and its associated positive eigenvalue $\lambda_k > 0$. Therefore, $\boldsymbol{u}_k > 0$ is also the unique(!) minimal eigenvector of $X_k$. Since $M \geq 2$, there exists a maximal eigenspace, in which any maximal eigenvector $\boldsymbol{y}_k$ satisfies $\boldsymbol{y}_k^\top \boldsymbol{u}_k = 0$. By Lemma 16, the theorem holds. $\square$

**Lemma 15.** *If the receptive field $R_k$ satisfies Assumption 1, and the collection of $N$ vectors $\{\boldsymbol{x}_k[i]\}_{i=1}^N$ contains all $M$ modes, then all off-diagonal elements of $\mathbb{C}_\alpha[\boldsymbol{x}_k, \boldsymbol{x}_k]$ are negative.*

*Proof.* We check every entry of $X_k := \mathbb{C}_\alpha[\boldsymbol{x}_k, \boldsymbol{x}_k]$. Let $\beta_i := \sum_{j \neq i} \alpha_{ij}$. Note that for off-diagnoal element $[X_k]_{ml}$ with $m \neq l$, we have:

$$[X_k]_{ml} = \sum_{ij} \alpha_{ij}(x_{km}[i]-x_{km}[j])(x_{kl}[i]-x_{kl}[j]) - \sum_i \beta_i(x_{km}[i]-x_{km}[i'])(x_{kl}[i]-x_{kl}[i']) \quad (87)$$

Let $A_m := \{i : x_{km}[i] > 0\}$ be the sample set in which the $m$-th component is strictly positive, and $A_m^c := \{1, 2, \ldots, N\} \backslash A_m$ its complement. By Assumption 1(one-hotness), if $i \in A_m$ then $i \in A_{m'}^c$ for any $m' \neq m$.

Now we consider several cases for sample $i$ and $j$:

**Case 1,** $i, j \in A_m$. Then $i, j \in A_l^c$ for $l \neq m$. This means that $x_{kl}[i] - x_{kl}[j] = 0$.

**Case 2,** $i, j \in A_m^c$. Then $x_{km}[i] - x_{km}[j] = 0$.

**Case 3,** $i \in A_m$ **and** $j \in A_m^c$. Since $j \in A_m^c$, we have $x_{km}[i] - x_{km}[j] = x_{km}[i] > 0$. On the other hand, since $i \in A_m$, $i \in A_l^c$, we have $x_{kl}[i] - x_{kl}[j] = -x_{kl}[j] \leq 0$. Therefore, $(x_{km}[i] - x_{km}[j])(x_{kl}[i] - x_{kl}[j]) \leq 0$.

**Case 4.** $i \in A_m^c$ **and** $j \in A_m$. This is similar to Case 3.

Putting them all together, since $\alpha_{ij} > 0$, we know that

$$\sum_{ij} \alpha_{ij}(x_{km}[i] - x_{km}[j])(x_{kl}[i] - x_{kl}[j]) \leq 0 \quad (88)$$

Furthermore, it is strictly negative since for $i \in A_m$ and $j \in A_l$, we have

$$(x_{km}[i] - x_{km}[j])(x_{kl}[i] - x_{kl}[j]) = -x_{km}[i]x_{kl}[j] < 0 \quad (89)$$

By our assumption that the $N$ vectors $\{\boldsymbol{x}_k[i]\}_{i=1}^N$ contains all $M$ modes, both $A_m$ and $A_l$ are not empty so this is achievable.

For the second summation, by Assumption 1(Augmentation), either $i, i' \in A_m$ or $i, i' \in A_m^c$, it is always zero for $m \neq l$. $\square$

**Lemma 16.** *If $\boldsymbol{v} > 0$ is an all positive $d$-dimensional vector, $\boldsymbol{u}^\top \boldsymbol{v} = 0$, then*

$$\min_m u_m \leq -\frac{\min_m v_m}{d-k} \frac{\|\boldsymbol{u}\|_\infty}{\|\boldsymbol{v}\|_\infty} \quad (90)$$

*where $k$ is the number of nonnegative entries in $\boldsymbol{u}$.*

*Proof.* Let $m_0 := \arg\max_m |u_m|$. If $u_{m_0} = -\|\boldsymbol{u}\|_\infty = \min_m u_m$ then we have proven the theorem. Otherwise $u_0 := u_{m_0} \geq 0$. $u_{m_0}$ is the largest entry of $\{u_m\}$.

Since $\min_m u_m < 0$, by Rearrangement inequality we have:

$$0 = \boldsymbol{u}^\top \boldsymbol{v} = \sum_m u_m v_m \geq \left(\min_m v_m\right) u_0 + (d-k)\left(\max_m v_m\right)\left(\min_m u_m\right) \tag{91}$$

The conclusion follows. $\qquad\square$