# OpenReview forum: "Understanding Deep Contrastive Learning via Coordinate-wise Optimization"
_NeurIPS.cc/2022/Conference — NeurIPS 2022 Accept_

### Official Review · Reviewer_2mJ8 · 2022-07-10

**Rating:** 6
**Confidence:** 3
**Soundness:** 4 excellent
**Presentation:** 4 excellent
**Contribution:** 3 good

**Summary:**

In this work, authors show that contrastive learning (CL) is actually under a broad family of loss functions, and propose a unified formulation called \alphaCL. This formulation unifies not only various existing contrastive losses, but also is able to extrapolate to give novel contrastive losses beyond popular existing ones, providing a guide for designing new CL loss.

**Questions:**

Question:

1.	I see authors analyze the representation of CL based on two-layer ReLU network in section 5. We know that Relu is local linear function so that it provides some advantages for analysis. How about other functions such as leaky relu?

2.	In the experiments, authors propose different formulations for the selection of r. Could you tell readers how you design them, or which should they focus on if they want to design new loss. Such discussions may help readers to design their own losses based on their problems with your theorys.


**Limitations:**

Yes

**Strengths And Weaknesses:**

Strength:

1.	Although lots of CL loss has been proposed in the past few years motivated by different points, there is no work studying the interactions among them. This work is the first one, to my best knowledge, to reveal the connections or relations among these existing popular CL losses, with efficient proofs and clear descriptions.

2.	Up to now, the research of CL still remains an open problem what features does CL learn. Especially when we use black-box deep neural networks, it is difficult for us to understand the learned features. Based on the analysis of proposed general CL loss in this work, authors presents some special case to help up understand the representation learning by CL.

Weakness:

1.	Current analysis and discussion are still based on some constrained assumptions, such as assumption 1 in the paper. And the understanding for the learned features of CL is still for some easy special case.

2.	The experimental results are very easy but with lots of theoretical analysis in the main paper. For example, the quantitative results are obtained only on CIFAR-10 and STL-10, which is a very small dataset and easy task. Besides, authors just select ResNet-18 as the backbone. Readers may ask how about applying the new formulation in the big dataset such as Imagenet? How about the performance if we select different backbones? How about applying the new proposed formulations for other CL tasks such as graph learning? Moreover, it is better for authors to provide some visualized results to help readers understand their theories.

---

> ### Author Response · Authors · 2022-08-02
> **Thanks for your comments!**
>
> Please check the overall rebuttal post that addresses common issues.
>
> ## Analysis of 2-layer ReLU networks extended to LeakyReLU network.
> For now it would require non-trivial efforts to extend it to LeakyReLU. We leave it for future work.
>
> ## Visualization of how the theory works.
> Due to the space limit, in the main text of this paper, we focus on formal theorem statements. In our next revision, we will provide more visualization if page limit allows. The intuition is that the min players $\alpha$ dynamically pick the "hard negative pairs" to focus on in CL training, and picking the right pairs will accelerate the training process.

---

> ### Author Response · Authors · 2022-08-08
> **Please let us know if you have more questions / concerns.**
>
> Dear reviewer,
>
> Please let us know if you have more questions / concerns. Thanks!

---

### Official Review · Reviewer_pNXS · 2022-07-11

**Rating:** 8
**Confidence:** 5
**Soundness:** 4 excellent
**Presentation:** 4 excellent
**Contribution:** 4 excellent

**Summary:**

In this paper, the authors provide a different perspective of InfoNCE with a theoretical analysis. A broad family of loss functions could be represented by a coordinate-wise optimization on the network parameter and pairwise importance. The authors connect the relationships between PCA and InfoNCE. Based on this point, the authors demonstrate optimizing the network parameter equals recovering the optimal PCA solution.

**Questions:**

N/A

**Strengths And Weaknesses:**

Strengths
1. This paper, although very theoretical, is still very easy to follow. I really enjoyed reading this paper.
2. The novelty is undoubted. Although the relationship between contrastive learning and PCA is not the first time to propose. This paper gives a very interesting point about the min player and the max player. It also addresses the how the temperature parameter influences the hard samples mining.
3. This paper gives a much formal reformulation to demonstrate the max player is equivalent to Principal Component Analysis (PCA) for deep linear network, and almost all local minima are global and rank-1, recovering optimal PCA solutions.
4. The two-layer MLP based analysis provides a good reason for the dimensional collapse in InfoNCE and other methods.

Weakness:
The relationship between contrastive learning and PCA is not the first time to propose. In contrastive laplacian eigenmaps (Neurips2021), the random regular graph is used. It is a Laplacian form of PCA.

---

> ### Author Response · Authors · 2022-08-02
> **Thanks for your comments!**
>
> We really appreciate your encouraging comments! Please check the overall rebuttal post that addresses common issues.

---

### Official Review · Reviewer_n1zS · 2022-07-12

**Rating:** 7
**Confidence:** 4
**Soundness:** 3 good
**Presentation:** 4 excellent
**Contribution:** 3 good

**Summary:**

This paper study a general class of loss function for contrastive learning that contains existing losses as special cases, which is formulated as a coordinate-wise optimization. The authors further proved analyses on the max player on deep linear networks and 2-NN.

**Questions:**

Please see the "Strengths and Weaknesses" section.

**Limitations:**

As discussed in the "Strengths and Weaknesses" section, the core analyses on 2-NN relies on a strong Assumption 1, which needs remedy.

**Strengths And Weaknesses:**

Pros:
+ The problem itself is relevant and important for many practitioners in the community.
+ The methodology is novel and general; the perspective of coordinate-wise optimization is interesting and the analyses are sound.
+ The paper is well-written and easy to follow.

Cons:
- The experiments is a bit weak: CIFAR-10 and STL-10 are relatively simple datasets. Can the authors also consider more involved datasets such as CIFAR-100?
- Figure 2 does not have confidence bands (although I noted Table 1 does report standard deviations).
- Assumption 1 seems a bit restrictive. It might be not satisfied by real-world (even simple ones), which makes Theorem 5 and 6 less interesting.

Summary: although the theoretical analyses on the 2-NN case relies on a strong assumption, I think the problem formulation and the perspective of coordinate optimization are useful to the community. I recommend the authors (1) add experiments on larger real datasets, and add confidence bands in the figures; (2) empirically verify whether Assumption 1 holds; or construct synthetic datasets based on Assumption 1 and baselining it over other methods.

---

> ### Author Response · Authors · 2022-08-02
> **Thanks for your comments!**
>
> Please check the overall rebuttal post that addresses common issues.
>
> ## Confidence band in Figure 2
> We will add confidence bands in our next revision.

---

> ### Author Response · Authors · 2022-08-08
> **Please let us know if you have more questions / concerns.**
>
> Dear reviewer,
>
> Please let us know if you have more questions / concerns. Thanks!

---

> > ### Comment · Reviewer_n1zS · 2022-08-08
> > **Thank you for your response**
> >
> > Thank you for your response and clarifying my questions. I am satisfied with the added experimental results and don't have more questions now.

---

### Author Response · Authors · 2022-08-02
**Rebuttal**

We thank the reviewers for their insightful and encouraging comments!

All reviewers agree that the proposed $\alpha$-CL framework is novel and can lead to a new landscape of loss design for contrastive learning. We will address the reviewer's detailed concerns below.

## Experiments on different backbones, datasets and batchsize (reviewer **n1zS** and **2mJ8**)

We now provide more experiments over different backbone (ResNet50), more complicated datasets (CIFAR100) and different batchsize (Batchsize=256). Overall, we see consistent gains of $\alpha$-CL over InfoNCE in the early stage of the training (e.g., 1-2 point of absolute percentage gain) and comparable performance at 500 epoch. Moreover, these performance improvements can be achieved with the same direct $\alpha$-mapping across different settings, i.e.,

$$\alpha_{ij} = \exp(-d_{ij}^p / \tau)$$

for $p=4$ (line 304 in the main text). Since this is a single point of design choice, more gains are possible if searching over the design space of the min player, which we leave for future work. Applying $\alpha$-CL to ImageNet and graph learning are interesting to explore later.

Here are details (we report mean/std for 5 trials):

#### CIFAR100 with ResNet18 backbone. Learning rate=0.01, batchsize=128

|             | 100 Epoch      | 300 Epoch      | 500 Epoch      |
|-------------|----------------|----------------|----------------|
| InfoNCE     | 55.696 ± 0.368 | 59.706 ± 0.360 | 59.892 ± 0.340 |
| $\alpha$-CL | **57.144** ± 0.150 | **60.110** ± 0.187 | **60.330** ± 0.194 |

#### ResNet50 backbone. Learning rate=0.001, batchsize=128

| Dataset  | Methods     | 100 Epoch      | 300 Epoch      | 500 Epoch      |
|----------|-------------|----------------|----------------|----------------|
| CIFAR10  | InfoNCE     | 86.388 ± 0.157 | 89.974 ± 0.138 | 90.194 ± 0.232 |
|          | $\alpha$-CL | **87.406** ± 0.227 | **90.228** ± 0.185 | **90.366** ± 0.209 |
| CIFAR100 | InfoNCE     | 60.162 ± 0.482 | 65.400 ± 0.310 | 65.532 ± 0.297 |
|          | $\alpha$-CL | **62.650** ± 0.181 | **65.630** ± 0.263 | **65.636** ± 0.269 |
| STL10    | InfoNCE     | 81.635 ± 0.244 | 86.570 ± 0.174 | **87.900** ± 0.222 |
|          | $\alpha$-CL | **82.850** ± 0.171 | **86.870** ± 0.178 | 87.653 ± 0.175 |

#### ResNet50 backbone. Learning rate=0.001, batchsize=256

| Dataset  | Methods     | 100 Epoch      | 300 Epoch      | 500 Epoch      |
|----------|-------------|----------------|----------------|----------------|
| CIFAR10  | InfoNCE     | 86.836 ± 0.255 | 89.188 ± 0.150 | **91.070** ± 0.122 |
|          | $\alpha$-CL | **87.740** ± 0.280 | **89.762** ± 0.259 | 91.064 ± 0.094 |
| CIFAR100 | InfoNCE     | 60.704 ± 0.402 | 64.216 ± 0.186 | **66.844** ± 0.163 |
|          | $\alpha$-CL | **63.284** ± 0.313 | **65.708** ± 0.198 | 66.726 ± 0.133 |
| STL10    | InfoNCE     | 82.085 ± 0.310 | 86.955 ± 0.188 | 87.307 ± 0.165 |
|          | $\alpha$-CL | **83.003** ± 0.275 | **87.345** ± 0.276 | **87.632** ± 0.294 |

## Connection between contrastive learning and PCA (reviewer **pNXS**)
We will update to reference recent papers that relate CL with PCA (including COLES mentioned by reviewer **pNXS**), and how our work relates to them. The main difference here is that our work incorporates the analysis of loss function with that of network architecture, while previous works focus on analysis of loss function and treat the network as a blackbox.

## Intuition of how to pick the right min player (reviewer **2mJ8**)
In this paper, we open up a novel avenue for CL loss design and are just scratching the surface in designing the min player. The role of thumb is to have min players really focusing on hard negative pairs within the batch, rather than putting uniform weights. In fact, compared to vanilla InfoNCE, the proposed direct-$\alpha$ indeed focuses more on “hard negative pairs” (by having large exponent, $p=4$ versus $p=2$), whose distance is small in representation space. On the other hand, too much focus on such pairs can lead to worse performance, as shown below:

### Ablation study on different alpha_exponent $p$ in STL10 (line 304)

| $p=2$          | $p=4$          | $p=6$          | $p=8$          | $p=10$         |
|----------------|----------------|----------------|----------------|----------------|
| 83.737 ± 0.184 | 84.057 ± 0.240 | **84.083** ± 0.416 | 83.910 ± 0.279 | 83.562 ± 0.131 |

A principled way of min player design is left for future work.

## Assumption 1 is a bit restrictive
We acknowledge that Assumption 1 can be idealistic for the input signal, which enables us to analyze some interesting properties of 2-layer ReLU networks with CL loss, as a first step to push forward this hard problem. As mentioned by **pNXS**, this provides illustrative examples showing that dimension collapsing can still happen even in nonlinear cases. Finding weaker assumptions that lead to interesting theoretical results is highly nontrivial and we leave it for future work.

---

### Meta-Review · Area_Chair_8ZHK · 2022-08-26

**Recommendation:** Accept
**Confidence:** Certain

**Metareview:**

This paper offers an interesting perspective on the min player and the max player for contrastive learning. It also addresses how the temperature parameter influences hard sample mining. The meta-reviewer recommends acceptance of the paper as a poster.

**Award:**

No

---

### Decision · Program_Chairs · 2022-09-14

Accept